# Cannabinoid CB$_2$ receptor ligand profiling reveals biased signalling and off-target activity

Marjolein Soethoudt[1,2,*], Uwe Grether[3,*], Jürgen Fingerle[4], Travis W. Grim[5], Filomena Fezza[6,7], Luciano de Petrocellis[8], Christoph Ullmer[3], Benno Rothenhäusler[3], Camille Perret[3], Noortje van Gils[2], David Finlay[9], Christa MacDonald[9], Andrea Chicca[10], Marianela Dalghi Gens[10], Jordyn Stuart[11], Henk de Vries[2], Nicolina Mastrangelo[12], Lizi Xia[2], Georgios Alachouzos[1], Marc P. Baggelaar[1], Andrea Martella[1,2], Elliot D. Mock[1], Hui Deng[1], Laura H. Heitman[2,**], Mark Connor[11,**], Vincenzo Di Marzo[8,**], Jürg Gertsch[10,**], Aron H. Lichtman[5,**], Mauro Maccarrone[7,12,**], Pal Pacher[13], Michelle Glass[9] & Mario van der Stelt[1]

The cannabinoid CB$_2$ receptor (CB$_2$R) represents a promising therapeutic target for various forms of tissue injury and inflammatory diseases. Although numerous compounds have been developed and widely used to target CB$_2$R, their selectivity, molecular mode of action and pharmacokinetic properties have been poorly characterized. Here we report the most extensive characterization of the molecular pharmacology of the most widely used CB$_2$R ligands to date. In a collaborative effort between multiple academic and industry laboratories, we identify marked differences in the ability of certain agonists to activate distinct signalling pathways and to cause off-target effects. We reach a consensus that HU910, HU308 and JWH133 are the recommended selective CB$_2$R agonists to study the role of CB$_2$R in biological and disease processes. We believe that our unique approach would be highly suitable for the characterization of other therapeutic targets in drug discovery research.

[1] Department of Molecular Physiology, Leiden Institute of Chemistry, Leiden University, Einsteinweg 55, Leiden 2333 CC, The Netherlands. [2] Department of Medicinal Chemistry, Leiden Academic Centre for Drug Research, Leiden University, Einsteinweg 55, Leiden 2333 CC, The Netherlands. [3] Roche Innovation Center Basel, F. Hoffman-La Roche Ltd., Grenzachterstrasse 124, Basel 4070, Switzerland. [4] Department of Biochemistry, NMI, University Tübingen, Markwiesenstrasse 55, Reutlingen 72770, Germany. [5] Department of Pharmacology and Toxicology, 1220 East Broad Street, PO Box 980613, Richmond, Virginia 23298-0613, USA. [6] Department of Experimental Medicine and Surgery, Tor Vergata University of Rome, Via Montpellier 1, Rome 00133, Italy. [7] European Center for Brain Research/IRCCS Santa Lucia Foundation, via del Fosso del Fiorano 65, Rome 00143, Italy. [8] Endocannabinoid Research Group, Institute of Biomolecular Chemistry, C.N.R., Via Campi Flegrei 34, Comprensorio Olivetti, Pozzuoli 80078, Italy. [9] Department of Pharmacology and Clinical Pharmacology, Faculty of Medical and Health Sciences, University of Auckland, 85 Park road, Grafton, Auckland 1023, New Zealand. [10] Institute of Biochemistry and Molecular Medicine, University of Bern, Bühlstrasse 28, Bern CH-3012, Switzerland. [11] Department of Biomedical Sciences, Faculty of Medicine and Health Sciences, Macquarie University, North Ryde, New South Wales 2109, Australia. [12] Department of Medicine, Campus Bio-Medico University of Rome, Via Alvaro del Portillo 21, Rome 00128, Italy. [13] Laboratory of Cardiovascular Physiology and Tissue Injury, National Institute of Health/NIAAA, 5625 Fishers Lane, Rockville, Maryland 20852, USA. * These authors contributed equally to this work. ** These authors jointly supervised this work. Correspondence and requests for materials should be addressed to P.P. (email: pacher@mail.nih.gov) or to M.G. (email: m.glass@auckland.ac.nz) or to M.v.d.S. (email: m.van.der.stelt@chem.leidenuniv.nl).

Target validation is an essential element of pharmacological research and drug discovery[1]. Pharmacological intervention using chemical probes provides a powerful means to assess the temporal consequences of acute modulation of protein function under both physiological and pathological conditions[1]. High selectivity and well-defined molecular mode of action of chemical probes are essential to translate the preclinical studies on non-human species to the patient. This type of information is, however, often lacking and reproducibility across different laboratories is sometimes difficult to obtain.

There is a great interest in the development of selective type-2 cannabinoid receptor ($CB_2R$) agonists as potential drug candidates for various pathophysiological conditions[2], which include chronic and inflammatory pain[3,4], pruritus[5], diabetic neuropathy and nephropathy[6,7], liver cirrhosis[8], and protective effects after ischaemic-reperfusion injury[9–12]. $CB_2R$ belongs to the cannabinoid receptor family of G protein-coupled receptors, which also includes type-1 cannabinoid receptor ($CB_1R$). Both CBRs are the biological target of $\Delta^9$-tetrahydrocannabinol ($\Delta^9$-THC), the main psychoactive component in cannabis[13,14]. $CB_1R$ and $CB_2R$ share an overall homology of 44%, but the 7-transmembrane spanning region, which contains the ligand-binding domain, exhibits 68% similarity[15]. $CB_2R$ is predominantly expressed on immune cells and its expression level is believed to increase in tissues upon pathological stimuli[2], whereas the $CB_1R$ is highly expressed in the brain[16]. Both receptors couple to $G_{i/o}$ proteins and modulate various intracellular signal transduction pathways, such as inhibition of cAMP-production, activation of pERK and G protein-coupled Inward Rectifying $K^+$-channels (GIRKs) and recruitment of β-arrestin to the receptor[17–19]. It is currently unknown which signal transduction pathways (or combinations thereof) are relevant for therapeutic purposes. In addition, some compounds may act as biased and/or protean agonists[18,19], and remarkable differences between rodent and human receptor orthologues have been noted, which are complicating the translation of results from preclinical animal models to human trials.

Different chemical classes have been described as CBR ligands (for example, mixed CBR agonists: $\Delta^9$-THC (henceforth referred to as THC), CP55940, WIN55212-2, HU210, and the endogenous ligands 2-arachidonoyl glycerol (2-AG) and anandamide (AEA, N-arachidonoylethanolamine); $CB_1R$ antagonists: SR141716A (rimonabant), and AM251; $CB_2R$ agonists: HU308, HU910, Gp-1a, JWH015, JWH133 and AM1241; and $CB_2R$ antagonists: AM630 and SR144528; see Supplementary Information, Supplementary Fig. 1 for structures)[2,20]. These ligands are used to explore CBR biology and to obtain preclinical target validation of the CBR subtypes[21]. The high homology between the ligand binding domains of the two receptors and the overall higher tissue expression of $CB_1R$ pose challenges to develop selective ligands that target only $CB_2R$. Yet, high selectivity is required to determine the exact role of each receptor in various (patho)physiological processes and to avoid $CB_1R$-mediated (psychotropic) side effects caused by THC and other $CB_1R$ ligands. The need for highly selective $CB_2$ ligands is exemplified by the scientific dispute whether the $CB_2R$ plays an important role in normal brain function or not. This whole avenue of research is currently being hampered by possible bias of using non-selective pharmacological, immunological and genetic tools and has delayed the development of novel $CB_2R$-based drugs[22,23].

Currently, most ligands are only characterized in a binding assay and/or in a limited set of functional assays using recombinant human receptors. The results are scattered among various publications and are derived from different experimental settings, which may have led to apparent contradictory results[23]. Conflicting results from in vivo models that employ some of the above-mentioned ligands have also been described in the literature (for a review see refs 2,24). Often, information about potential off-targets and pharmacokinetics of ligands is also lacking[19], which has complicated the comparison and interpretation of the data and led to confusion about which are the preferred ligands to be used for in vivo experiments aimed at validating the $CB_2$ receptor as a therapeutic target. Unfortunately this situation, which has resulted in a loss of resources and unnecessary use of animals, is not unique to the $CB_2$ receptor field. The US National Institutes of Health (NIH) shares these concerns from many scientists about the reproducibility issues in biomedical research and required action to counter this problem[25]. To improve target validation and to guide the selection of the best ligand for preclinical studies, a fully detailed profile of the current 'gold standard' ligands is needed.

To provide important guidance for the field and to address potential species-dependent differences, we comprehensively profiled the most widely used $CB_2R$ ligands. In several independent academic and industry laboratories we investigated receptor binding of both human and mouse $CB_2R$, as well as multiple signal transduction pathways (GTPγS, cAMP, β-AR, pERK and GIRK). Selectivity of the ligands was determined towards a customized panel of proteins associated with cannabinoid ligand pharmacology, which includes the $CB_1R$ and the major proteins of the endocannabinoid system: N-acyl ethanolamines biosynthesizing enzyme NAPE-PLD and AEA hydrolysing enzyme FAAH; 2-AG biosynthesizing enzyme DAGL and hydrolysing enzymes MAGL, ABHD6 and ABHD12, as well as towards the putative endocannabinoid transporters; AEA and 2-AG-binding transient receptor potential (TRP)-channels (TRPV1–4, TRPM8 and TRPA1) (for a review see ref. 26). In addition, off-target activity on GPR55, a receptor that binds CBR-type ligands, and on COX-2, which oxygenates AEA and 2-AG, was also determined. Determination of the selectivity of $CB_2R$ ligands over these other proteins and processes involved in the endocannabinoid system, as well as over the TRP channels (which are involved in similar biological processes as the CBRs) is essential for the development of selective $CB_2R$ ligands and to avoid complications in the interpretation of the in vivo results obtained with these compounds.

To assess which ligands are best suited for in vivo studies, all 18 compounds are profiled for their physico-chemical properties, in vitro absorption distribution metabolism and excretion (ADME) and pharmacokinetic parameters and cross-reactivity in the CEREP panel of 64 common off-targets. Commonly used non-selective ligands, including $\Delta^9$-THC and the endocannabinoids 2-AG and anandamide are also tested in vitro. All ligands are high-quality grade material, provided to each laboratory by the industry collaborator. The top three candidate $CB_2R$ agonists are further investigated at high doses in vivo to infer potential interactions with CNS $CB_1R$. All data together results in the largest data set generated so far under the same experimental conditions for all cannabinoid receptor ligands, leading to a consensus that HU910, HU308 and JWH133 possess the best $CB_2R$ agonist profiles among the ligands tested on the basis of selectivity, balanced signalling, pharmacokinetic profile and off-target activity, and may be considered 'gold standards' for $CB_2R$ validation studies in mice.

## Results

**Physico-chemical properties.** The physico-chemical properties of the 18 compounds tested are listed in Supplementary Table 1 of the Supplementary Information. Molecular weights span

a range from $312 \, \mathrm{g \, mol^{-1}}$ for JWH133 up to $555 \, \mathrm{g \, mol^{-1}}$ for AM251 and the polar surface area values are overall very low (8 Å for JWH133 up to 63 Å for (S)-AM1241), due to a low number of heteroatoms present in the ligands. Importantly, all CBR ligands are very lipophilic molecules, which negatively affect their solubility, ADME-properties and off-target profile. Even the lowest lipophilicity value (clogP), calculated to be 4.9 for WIN55212-2, is relatively high. The most lipophilic CBR ligand is SR144528, which exhibits an extremely high clogP value of 9.2. Consequently, only CP55940 and (rac)-AM1241 were soluble in an aqueous phosphate buffer system (pH 6.5). Despite the fact that the membrane permeation coefficient (PAMPA) $P_{\mathrm{eff}}$ is low for several of the molecules, most compounds are expected to be able to cross biological barriers as high percentages of the substances were found in membranes.

**Affinity and selectivity in CBR binding studies.** To determine the affinity and selectivity of the 18 substances, we performed [³H]-CP55940 displacement assays using membrane fractions of CHO cells expressing recombinant human $CB_2R$ and $CB_1R$, in two independent laboratories. In addition, mouse brain and spleen were used as source of mouse $CB_1R$ and $CB_2R$, respectively.

Using the Pearson correlation analysis, we found a statistically significant correlation between the binding affinities between the different labs (Pearson coefficient: 0.9304 ($hCB_1R$), 0.6648 ($hCB_2R$) and 0.7720 ($mCB_2R$), see Supplementary Fig. 2). Figure 1 depicts the selectivity of the ligands for the $CB_2R$ versus $CB_1R$. We found that HU210 > CP55940, WIN55212-2 > $\Delta^9$-THC were the highest affinity non-selective human CBR ligands. Conversely, HU308, HU910 and JWH133 were the most selective human $CB_2R$ ligands (Supplementary Table 2), possessing 278-, 166- and 153-fold higher respective affinities for $CB_2R$ than for $CB_1R$. Notably, JWH015 and Gp-1a were less than 30-fold selective for $CB_2R$. Importantly, the binding selectivity of the ligands for mouse $CB_2R$ over mouse $CB_1R$ appeared to be greatly reduced for all ligands (<100 fold), except AM630 and SR144528, which are actually more selective

on $mCB_2R$ than on $hCB_2R$. The most selective agonists on $mCB_2R$ were (rac)-AM1241 (66-fold), JWH133 (40-fold) and Gp-1a (20-fold). As expected, AEA and 2-AG, the endogenous ligands of $CB_1R$ and $CB_2R$, were non-selective and showed moderate binding affinities towards both receptors (pKi ~7).

**Activity and selectivity of CBR signalling pathways.** To determine the functional activity and selectivity (towards $CB_2R$ over $CB_1R$) of the ligands we performed five different assays (GTPγS, cAMP, β-AR, pERK and GIRK) on both human $CB_2R$ and human $CB_1R$ (Supplementary Tables 3–7). All ligands were tested on cAMP signalling on both mouse CBRs and HU910, HU308 and JWH133 were tested on G-protein activation and β-arrestin recruitment on $mCB_2R$, to determine interspecies behaviour of the ligands. Efficacy of the ligands is normalized to the effect produced by CP55940 (10 µM) in all assays; however, it should be noted that efficacy is relative by definition, and is dependent on the reference ligand used as well as the assay conditions.

For both human and mouse $CB_2R$, the potency of the ligands correlated with their binding affinity in most assays, except for β-AR and GIRK signalling (Supplementary Fig. 3). Graphs showing the pEC50 values of the reference ligands for all assays are shown in Fig. 2a–d.

CP55940 and HU308 behaved as potent full agonists at $hCB_2R$ in the GTPγS assay (Supplementary Table 3), while WIN55212-2 acted as a partial agonist. HU910 behaved as a partial $CB_2R$ agonist as well, but was, together with HU308 and JWH133, the most selective for $CB_2R$ in this assay (185- and 193-fold, respectively). Of note, JWH133 was considered functionally inactive on $hCB_1R$, because its maximal effect was only 20% at 10 µM. On $mCB_2R$, both HU308 and JWH133 were full agonists, but HU910 remained partially active. The potency of all three ligands was similar for human and mouse receptors.

In contrast to previous reports[27,28], Gp-1a acted as an inverse agonist on $CB_2R$, but was inactive at $CB_1R$. Both THC and the endocannabinoids AEA and 2-AG acted as partial agonists on both receptors with similar potency.

In the cAMP assay (Supplementary Table 4) all $CB_2R$ agonists displayed higher selectivity (>1,000-fold) and higher efficacy, than in the GTPγS-assay, reflecting substantial signal amplification in this pathway. Only (rac)-AM1241 remained a partial agonist in the cAMP assay. Upon comparison of the efficacy of the ligands between species, in general it appears that many ligands on the mouse CBRs are partial agonists on cAMP, in contrast to the human CBRs. This difference in efficacy might be a result of a difference in CBR expression levels. Differences in expression levels may also account for the interspecies differences displayed by (rac)-AM1241, which was previously reported as a protean agonist (a protean agonist shows differences in signalling due to differences in experimental conditions, whereas a true biased agonist has signalling preference due to conformational changes of the receptor)[29]. HU910 was found to bind with similar affinity to both mouse CBRs, but was inactive on $mCB_1R$ in the cAMP assay. HU308, JWH015 and (rac)-AM1241 were the most selective agonists for $hCB_2R$ in this assay, whereas CP55940, WIN55212-2 and HU210 displayed the highest potency. HU910, HU308 and JWH133 were the most selective on the $mCB_2R$. Of note, AEA and 2-AG were relatively weak partial agonists, especially on human and mouse $CB_1R$ (pEC50 < 5.2 and Emax < 70%).

All ligands modulated β-AR recruitment (Supplementary Table 5) to the membrane in CHO cells expressing human $CB_2R$ or $CB_1R$. CP55940 was the most potent ligand in this assay, followed by WIN55212-2. CP55940 acted as full agonist at both receptors, but WIN55212-2 displayed partial agonism at

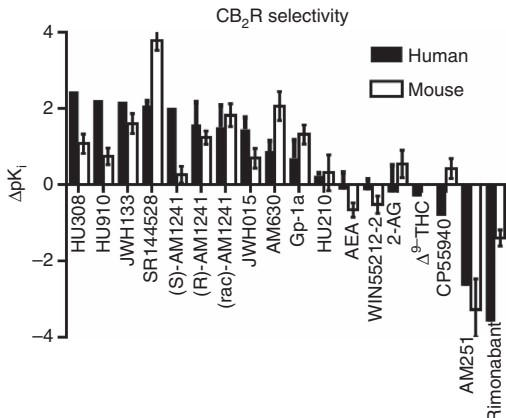

**Figure 1 | $CB_2R$ selectivity of cannabinoid reference ligands on mouse and human CBRs.** $CB_2R$ selectivity for all cannabinoid receptor reference ligands are presented as the difference in mean pKi values between $CB_2R$ and $CB_1R$ for both the human (black bars) and mouse (white bars) orthologues. From left to right: ligands with decreasing $CB_2R$ selectivity (from HU308 to Gp-1a), nonselective ligands (from HU210 to CP55940), ligands with $CB_1R$ selectivity (AM251 and SR141716A). pKi values are obtained from three independent experiments performed in duplicate. Error bars shown in the figure represent the s.e. of the mean.

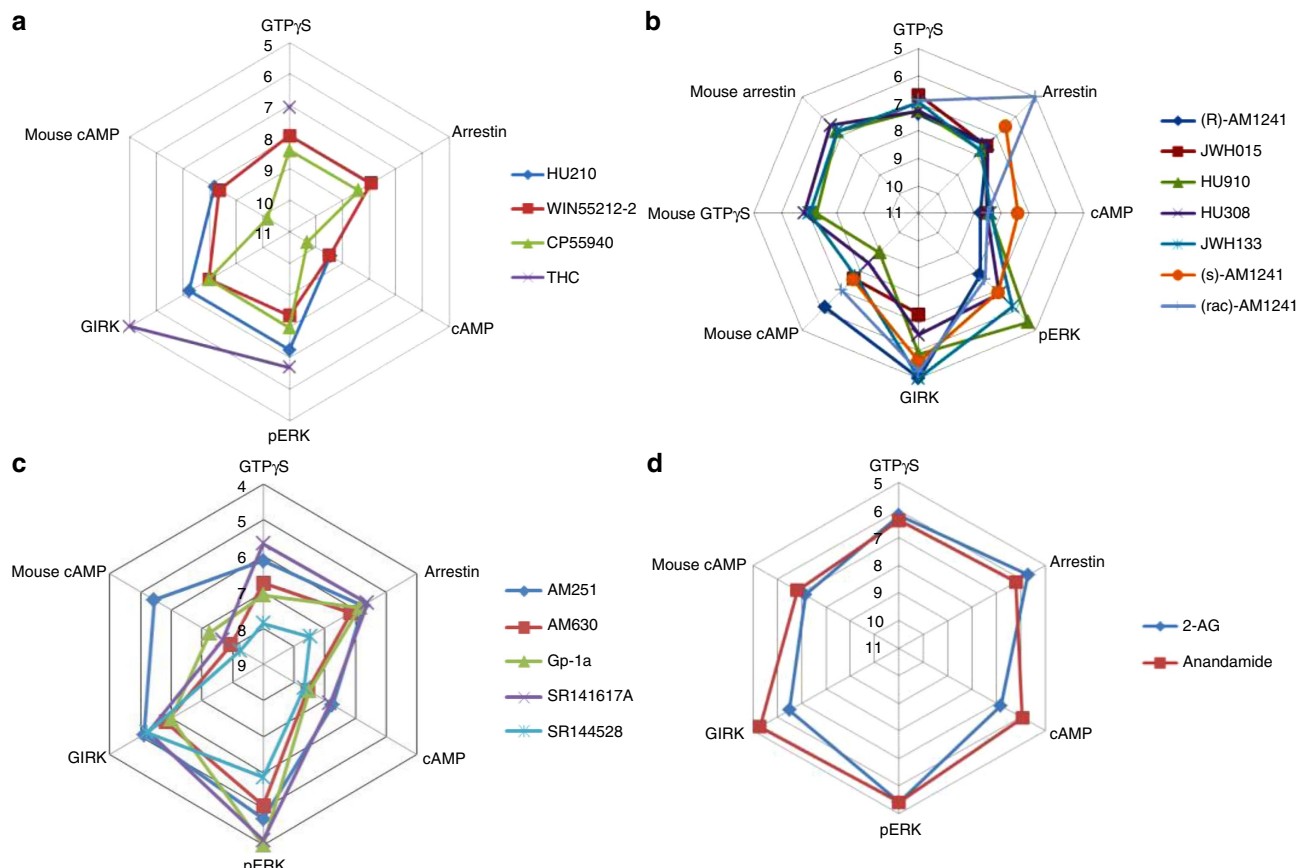

**Figure 2 | Radarplots of potencies of cannabinoid receptor reference ligands across different assays.** (**a**) non-selective CBR agonists, (**b**) selective CB$_2$R agonists, (**c**) antagonists, (**d**) endocannabinoids. These plots are a visualization of the differences in potency across different signalling pathways. All potency values are the mean of three independent experiments performed in duplicate.

CB$_2$R, as in the GTPγS assay. The other agonists, including the ligands JWH133, JWH015, HU308, HU910 (Emax 50–70%) and the endocannabinoids (Emax 40–80%), only partially recruited β-AR. The most selective CB$_2$R agonists were HU308, JWH133 and HU910, which were found to be 282-, 275- and 274-fold more potent for CB$_2$R than CB$_1$R. Of note, JWH133, HU308 and HU910 were all significantly less potent on mCB$_2$R in β-AR recruitment.

WIN55212-2 was a full agonist in the pERK assay (Supplementary Table 6) and demonstrated 86-fold selectivity for the human CB$_2$R, whereas CP55940 lacked selectivity in this assay. HU308 and JWH133 were potent and selective CB$_2$R full agonists, whereas Δ$^9$-THC and (Rac)-AM1241 acted as partial agonists on the pERK signalling cascade. Interestingly, HU910, AEA and 2-AG had low potency in this assay (pEC50 < 5.5), but HU910 and 2-AG acted as full agonists at high concentrations.

Most ligands appeared to be less potent and less CB$_2$R-selective in the GIRK assay (Supplementary Table 7). For example, neither JWH133 nor THC activated the GIRK pathway at all. JWH015 was the most selective agonist in this assay, followed by HU308. WIN55212-2 and CP55940 activated the GIRK channels with the highest potency, but as expected, both were highly potent and efficacious at CB$_1$R as well.

The high variability in potency and efficacy that the CB$_2$R agonists displayed across the different signalling pathways strongly suggests biased signalling. To quantify this ligand bias towards distinct signal transduction pathways, we performed operational analysis based on van der Westhuizen *et al.* (Supplementary Tables 8–11) (ref. 30). This analysis is based

on the operational model of Black and Leff[31], which calculates signal transduction strength on a given pathway, taking into account (a) the maximal effect of the system used, (b) the agonist concentration, (c) the agonist's maximum efficacy, (d) the ligand affinity for the receptor and (e) the transducer slope. In order to eliminate system and observation bias, such as the level of amplification between signalling pathways or assay sensitivity, we used CP55940 as a reference compound, because it was the only compound that behaved as a full agonist with comparable potency in all assays, except for the cAMP assay. The ΔΔlogR values resulting from this operational analysis are graphically shown in Fig. 3. The operational analysis on hCB$_2$R revealed that THC showed statistically significant bias towards pERK signalling compared to β-arrestin and GTPγS. In addition, THC did not activate GIRK, indicative of high bias against this pathway. (rac)-AM1241 was biased towards β-arrestin coupling and pERK signalling compared to GIRK channel activation. JWH133 was moderately biased towards β-arrestin compared to GIRK, whereas both WIN55212-2 and JWH015 showed preference for GIRK compared to cAMP signalling. AEA showed preference for pERK and GIRK signalling compared to cAMP, whereas 2-AG was significantly biased towards GIRK compared to G-protein signalling. Upon comparison between β-AR coupling and cAMP signalling, all ligands appear to be significantly biased. This observation is, however, confounded by the fact that CP55940, which is used as the reference ligand, has an exceptionally high potency in the cAMP assay compared to the other signal transduction pathways, and might be in fact biased itself towards the cAMP

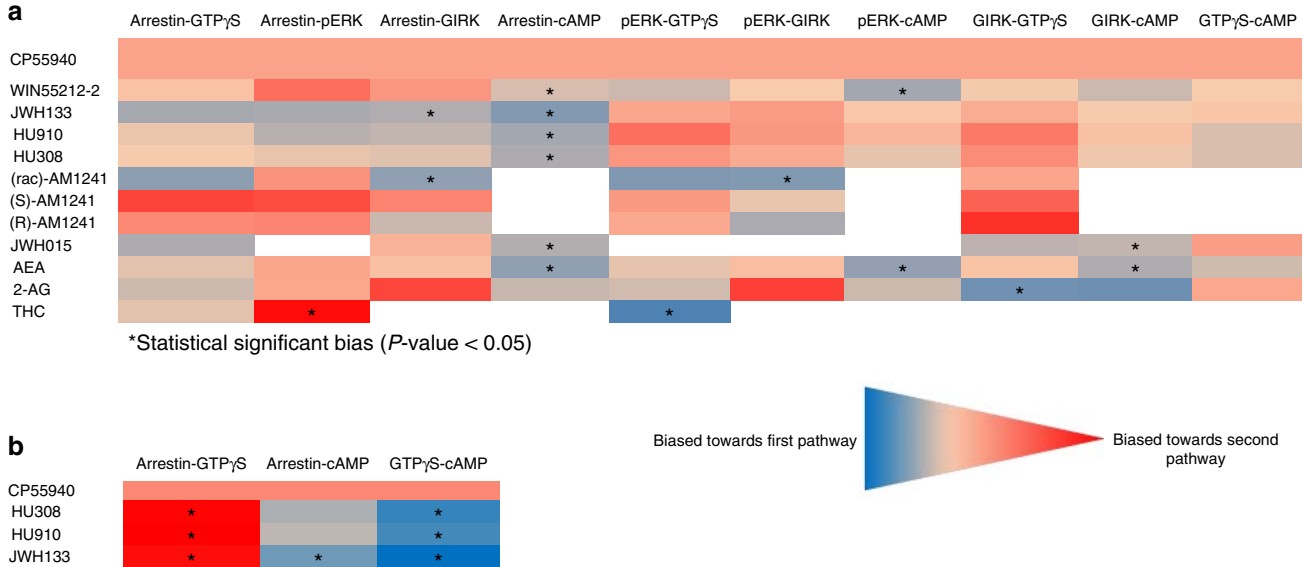

**Figure 3 | Heatmap of ΔΔlogR values resulting from the operational analysis.** The operational analysis was performed using the procedure described in the Supplementary Information, with data obtained from three independent experiments performed in duplicate. Dark blue indicates bias towards the first pathway, while dark red indicates bias towards the second pathway. White boxes are not determined. (**a**) The operational analysis on hCB$_2$R was done with all agonists, except HU210, which was tested in only two functional assays due to legal restrictions. Because of error propagation in the analysis, only a few were found to be statistically significant ($P < 0.05$). (**b**) The operational analysis on mCB$_2$R was done with the three most selective and most broadly active compounds on hCB$_2$R; HU308, HU910 and JWH133. Here, all three ligands showed statistically significant bias towards G-protein signalling. Statistics was performed with one-way ANOVA Holm-Sidak's multiple comparisons test using Graphpad Prism 6.

pathway. Of note, HU910 and HU308 were well-balanced ligands without significant bias towards any signal transduction pathway on hCB$_2$R. Of note, HU910, HU308 and JWH133 were significantly biased towards G-protein signalling over β-AR coupling and cAMP signalling on the mCB$_2$R highlighting a potentially important species difference.

**Off-target activity in the endocannabinoid system.** To rule out any indirect effects of the ligands on CB$_1$R and CB$_2$R, we investigated off-target activity on endocannabinoid-regulating enzymes, as well as their effects on AEA reuptake inhibition. None of the ligands showed any off-target activity on a panel of serine hydrolases, determined in a competitive activity-based protein profiling (ABBP) assay in mouse brain proteome up to a concentration of 10 μM (Supplementary Fig. 4A). In addition, none of the ligands showed any significant effect at a concentration of 10 μM when tested on NAPE-PLD, DAGL and MAGL-overexpressing cells (Supplementary Table 12). The compounds were also tested on FAAH activity using U937 cell homogenate at a concentration of 5 μM. Only AM251 and Gp-1a showed partial inhibition of FAAH activity (∼30–40%, Supplementary Fig. 4F).

In addition, none of the agonists had significant activity on ABHD6-, ABHD12- and COX2 activity up to a concentration of 5 μM for COX2 and 10 μM for the ABHDs (Supplementary Fig. 4B–E). In contrast, the antagonists SR141716A, AM251 and Gp-1a, which are the structurally most similar ligands in this panel of ligands, inhibited ABHD12 in the micromolar range with IC$_{50}$ values of 6.1, 1.6 and 0.8 μM, respectively (Supplementary Table 12). AM251 was the only compound of the ligands tested that had high efficacy on GPR55 in β-arrestin recruitment (82 ± 9%), albeit with low potency (pEC$_{50}$ = 5.49 ± 0.09, see Supplementary Table 13).

AEA reuptake inhibition was determined in three different human cell lines: monocyte-like U937 cells, mast cell-like HMC-1

cells and keratinocyte-like HaCaT cells. In U937 and HaCaT cells, some of the ligands possessed micromolar potency, including AM251, SR141716A, Gp-1a, HU308 and HU910 (Supplementary Table 14). In HMC-1 cells, which lack FAAH expression[32], all tested ligands, except SR141716A, were weakly active or inactive at a concentration of 5 μM, which indicates a potential role of FAAH in the inhibition of AEA uptake in U937 and HaCaT cells. In agreement with this, the most active AEA reuptake inhibitors AM251 and Gp-1a partially inhibited FAAH at 5 μM in U937 cell homogenate (Supplementary Fig. 4F). Of note, SR144528 was inactive up to a concentration of 10 μM in all cell lines, whereas SR141716A showed FAAH-independent micromolar effects on AEA reuptake in all cells.

**TRP-channels.** As AEA and 2-AG activate some TRP channels, these channels may be regarded as ionotropic cannabinoid receptors. Here, we tested our ligands on six different TRP channels (TRPV1–4, TRPA1 and TRPM8). We found that most TRP channels were activated by one or more ligands, apart from TRPA1 that was activated by all of them (Supplementary Table 15). HU308 was the most selective agonist that activated only TRPV1 and TRPA1 in the high micromolar range, whereas HU910 activated TRPV3 with submicromolar potency (0.12 ± 0.05 μM). However, to date, TRPV3-related effects and toxicity (hypothermia and reduced blood pressure) have not been observed after in vivo administration of HU910 (ref. 33).

JWH133 only activated TRPA1. Although its efficacy was fairly high (76.8 ± 3.8% activation), its potency was low (8.5 ± 2.3 μM). Whereas SR144528 did not target any of the TRP channels, AM630 was a full agonist at TRPA1 (118 ± 2%). Of note, SR141716A targeted three out of six TRP channels tested and, remarkably, nanomolar potency was observed at TRPM8.

**Off-target panel (CEREP panel).** The CEREP panel served to screen off-target activity on 64 proteins, which are associated with

common adverse side effects in humans. JWH133 was identified as the most selective ligand with no off-targets detected in this panel. The summary of all off-targets is shown in Supplementary Table 16, in Supplementary Table 20 all CEREP data are shown. In contrast, CP55940 was the most non-selective ligand of which we detected 17 off-targets with more than 50% inhibition at 10 μM. HU910 and HU308 hit nine and four off-targets in this panel, respectively. However, of the nine off-targets of HU910 in this panel only the dopamine uptake reporter displayed an $IC_{50}$ of $< 10 \mu M$ ($IC_{50} = 1.40 \mu M$). Therefore, these off-targets are not likely to be physiologically relevant at 10 μM. The $CB_2R$-selective antagonist SR144528 had only two off-targets. Of note, the adenosine $A_3$ receptor was the most common off-target (Supplementary Table 16). The physiological relevance of this observation is currently unclear, although it has previously been published that the endocannabinoid 2-AG has allosteric activity on this receptor[34].

**In vitro DMPK and pharmacokinetics**. The high overall lipophilicity (see above) may strongly influence other ADME properties. Metabolic stability in human and mouse microsomes and hepatocytes is low for many of the ligands as indicated by their high in vitro clearances (Supplementary Table 17). Some compounds, such as AEA, suffered from high chemical instability even in dimethylsulfoxide (DMSO) stock solution. For most ligands, except Gp-1a, microsomal clearances seem to over-predict the corresponding values in both human and mouse hepatocytes. The fraction unbound (Fu) was either very low or not measured for many compounds due to their very high lipophilicity. None of the molecules is a strong human or mouse P-gp substrate. In combination with their low polar surface areas, the ligands are, therefore, likely to reach the brain.

We determined the primary pharmacokinetic parameters of the compounds (Supplementary Table 18). A mixture containing 15% DMSO and 85% PEG400 was used as a vehicle to dissolve the ligands at $1–2 \, mg \, kg^{-1}$ for intravenous (i.v.) administration. In vivo clearances in mice were very high and span a range from the lowest value of $0.17 \, l \, h^{-1} \, kg^{-1}$ for HU910 up to $6.9 \, l \, h^{-1} \, kg^{-1}$ for (S)-AM1241. The volume of distribution was high ($> 3 \, l \, kg^{-1}$) for (S)-AM1241, AM630, Gp-1a and JWH015, moderate ($1–3 \, l \, kg^{-1}$) for AM251 and JWH133 and low ($< 1 \, l \, kg^{-1}$) for HU910 and SR144528. This resulted in the longest half-life for HU910 (7 h), whereas JWH133 had a half-life of only 1 h. As reasonable C0 values, extrapolated from Cmax values, can be reached for all compounds (see Supplementary Table 18), the rather short in vivo half-lives raise the possibility that published mouse in vivo efficacy data using these molecules might be rather C0 than AUC driven. Future experiments using accurate concentration/effects relationships might answer this question.

Following oral administration using aqueous microsuspensions in rodents, absorption was strongly influenced by the physico-chemical properties of the compounds, formulation and feed conditions of the animals (Supplementary Table 19). HU910 was suspended in ethanol/cremophor EL/0.9% NaCl (5/5/90% w/w) whereas HU308 and JWH133 were suspended in an aqueous gelatine/NaCl vehicle (7.5/0.62% w/w). Maximal plasma concentration peaked ~1 h after administration with Cmax ranging from 201 to 2070 ng ml$^{-1}$. Half-lifes were comparable to i.v. administration. Bioavailability was not calculated as these were separate experiments. Taken together these data suggest a wide variety of application format for in vivo experiments if care is taken on the formulation aspects. Determination of the plasma concentrations and pharmacokinetic behaviour seems however warranted.

**In vivo selectivity of HU308 HU910 and JWH133**. Finally, to determine whether the three most selective agonists, HU308, HU910 and JWH133, elicited cannabimimetic pharmacological effects in vivo, these compounds were tested in assays highly sensitive to $CB_1R$ activity (catalepsy, antinociception and hypothermia). HU210, a non-selective, highly potent $CB_1R/CB_2R$ agonist was used for comparison. In addition, as binding data had suggested affinity for $mCB_1R$ ($pKi = 6.14 \pm 0.13$), HU910 was tested for antagonistic effects in these in vivo measures, and compared with SR141716A.

As shown in Fig. 4, HU210 dose-dependently elicited catalepsy (Fig. 4a; $F_{(7,42)} = 60.7$, $P < 0.0001$ (Dunnett's test); $ED_{50}$ (95% CL) = 0.19 (0.14–0.27) $mg \, kg^{-1}$), antinociception (Fig. 4b; $F_{(7,35)} = 257.5$, $P < 0.0001$ (Dunnett's test); $ED_{50}$ (95% CL) = 0.41 (0.31–0.54) $mg \, kg^{-1}$) and hypothermia (Fig. 4c; $F_{(7,35)} = 97.7$; $P < 0.0001$ (Dunnett's test); $ED_{50}$ (95% CL) = 0.35 (0.30–0.41) $mg \, kg^{-1}$). In contrast, HU308, HU910 and JWH133 did not produce detectable catalepsy (Fig. 4a), antinociception (Fig. 4b) or hypothermia (Fig. 4c) within the dose range tested ($1$-$100 \, mg \, kg^{-1}$).

To test whether HU910 behaves as a $CB_1$ receptor antagonist in vivo, we tested whether $30 \, mg \, kg^{-1}$ HU910 would antagonize the pharmacological effects of $1.7 \, mg \, kg^{-1}$ HU210 (that is, an approximate $ED_{84}$ dose). Whereas SR141716A ($3 \, mg \, kg^{-1}$) significantly antagonized the cataleptic (Fig. 4d; $F_{(1,10)} = 46.7$, $P < 0.0001$ (Holm-Sidak's test), antinociceptive (Fig. 4e; $F_{(1,10)} = 39.7$, $P < 0.0001$ (Holm-Sidak's test)) and hypothermic (Fig. 4f; $F_{(1,10)} = 11.6$; $P < 0.01$ (Holm-Sidak's test)) effects of HU210, HU910 did not significantly reduce the magnitude of HU210-induced catalepsy (Fig. 4g; $P = 0.12$ (Holm-Sidak's test)), antinociception (Fig. 4h; $P = 0.19$ (Holm-Sidak's test)) or hypothermia (Fig. 4i; $P = 0.40$ (Holm-Sidak's test)). These results indicate that HU308, HU910 and JWH133 lack $CB_1R$ activity at relevant concentrations in vivo.

## Discussion

Drug discovery research has focused on the design and synthesis of selective cannabinoid $CB_2R$ agonists. Selective activation of this receptor has been associated with anti-inflammatory and tissue protective effects without inducing $CB_1R$-mediated psychoactive side effects. This concept has been supported by the use of $CB_2R$ knock-out mice showing enhanced pathology in disease models, such as heart, liver or kidney injury and inflammation. It is unclear why two different $CB_2R$ agonists lacked efficacy in phase 2 clinical pain trials[35,36], despite compelling proof-of-concept data obtained in preclinical studies. This lack of translation not only suggests possible deficiencies in the predicative utility of the preclinical models, but also that improved understanding of the molecular actions of $CB_2R$ agonists is needed. In addition, selective $CB_2R$ ligands are essential to determine whether $CB_2R$ has a physiologically relevant role in the normal brain function, which is currently under intense scientific debate[22]. To answer this question, a truly $CB_2R$-selective ligand is needed to avoid confusion caused by the use of non-selective cannabinoid ligands[23].

Here, we have comprehensively characterized a set of 18 CBR ligands for their physicochemical properties, in vitro molecular pharmacology, off-target profile and pharmacokinetics to guide the selection of the optimal ligands to perform preclinical proof-of-concept studies. An important finding of our study is that most agonists display reduced selectivity in binding affinity and functional efficacy on the mouse $CB_2R$ versus $CB_1R$ compared to the human orthologues while the antagonists display opposite behaviour. This observation may potentially be explained by the fact that the agonists stabilize different

receptor conformations than antagonists (inverse agonists) by interacting with different (species specific) amino acids in the binding pocket.

The reduced selectivity of the agonists for the $mCB_2R$ is a limitation that needs to be taken into account, especially when designing studies to investigate (neuro)inflammation in mice[23]. In contrast to previous reports that classify Gp-1a as a $CB_2R$ agonist[27,28], we found Gp-1a to be a functional $CB_2R$ and $CB_1R$ antagonist (inverse agonist) in all functional assays both on human and mouse orthologues.

Another important finding in the present study is the provocative evidence indicating biased signalling of $CB_2R$ agonists. For target validation it would be advisable to use a balanced ligand, instead of a strongly biased agonist, until it becomes clear that activation of a specific pathway is desired, because this may complicate the translation to the human situation. In our ligand set, THC, 2-AG and (rac)-AM1241 behaved as the most biased agonists on $hCB_2R$, with each stimulating their most preferred pathway >100-fold stronger than their least preferred pathway. Of note, 2-AG and AEA had distinct profiles in signalling pathway activation, which might open possibilities to explain ligand diversification of CBRs[37]. Importantly, we found that HU308 and HU910 showed differences in signalling preference between the human and

mouse $CB_2R$, being well-balanced agonists in all five signal transduction pathways on $hCB_2R$, but significantly biased on $mCB_2R$ towards G-protein activation compared to β-arrestin recruitment and cAMP signalling. The consequences of these interspecies differences in signalling preference for the translation of preclinical models to the clinic needs to be taken into account when testing novel drug candidates.

In addition, we found that SR144528 is a very effective antagonist of $CB_2R$-mediated modulation of cAMP signalling, but less so on other signal transduction pathways of the same receptor (GIRK and pERK). This signalling-specific inhibition should also be taken into account when studying $CB_2R$ agonists that preferably act through these mechanisms.

To determine off-target activity of the cannabinoid reference library, we tested them on a customized panel of proteins that are associated with cannabinoid ligand pharmacology, including GPR55, proteins involved in the biosynthesis and metabolism of endocannabinoids (DAGL-α, NAPE-PLD, MAGL, ABHD6, ABHD12, COX-2, FAAH and endocannabinoid transporter activity) and the TRP ion channel family. In combination with the additional off-target data we collected using the CEREP panel, we found that most ligands displayed a rich poly-pharmacology (see for a summary of all off-targets per compound Fig. 5 and Supplementary Table 13). Remarkably, the highly

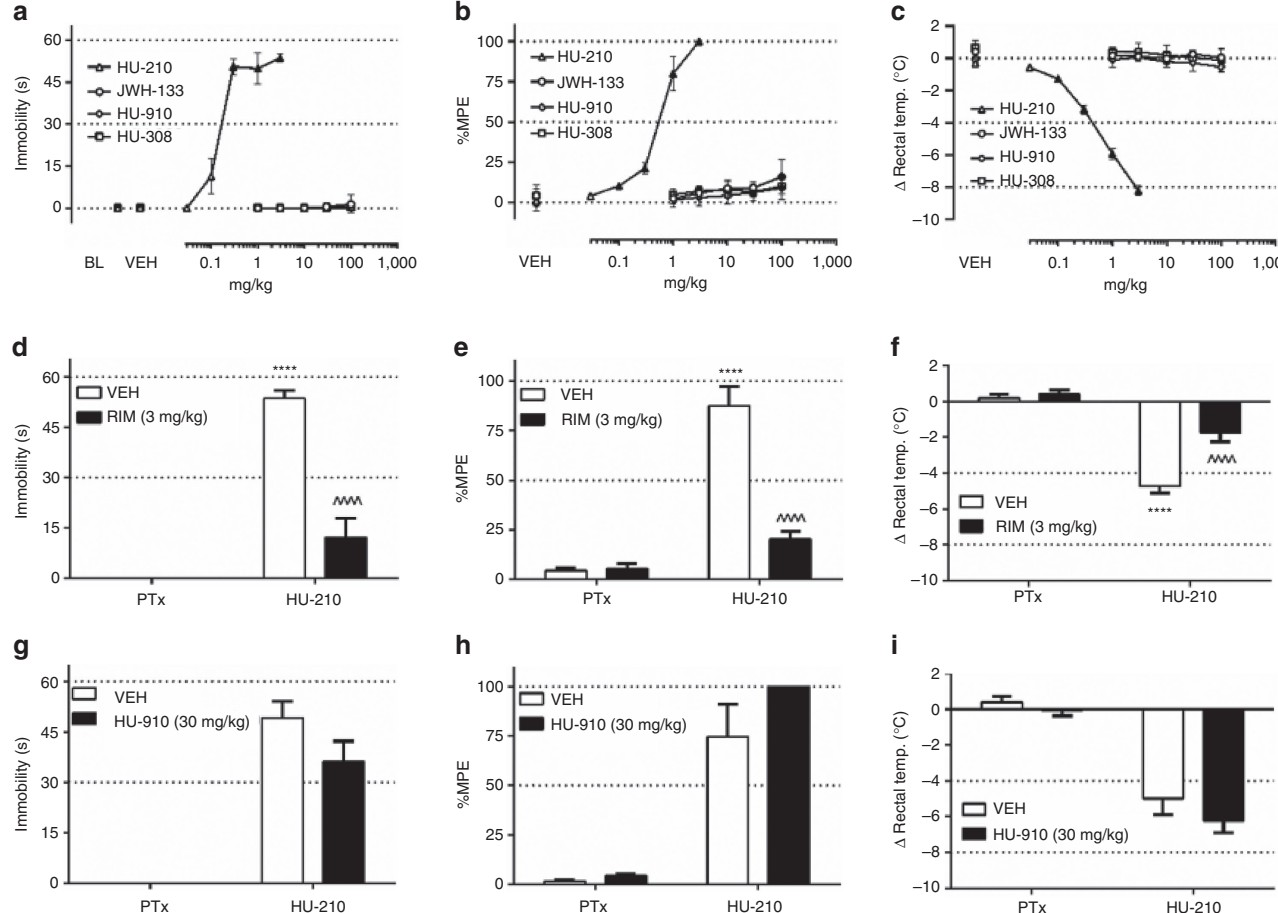

**Figure 4 | Comparison of *in vivo* cannabimimetic pharmacological effects.** HU210 dose-dependently elicited catalepsy (**a**), antinociception (**b**) and hypothermia (**c**), but HU308, HU910 and JWH133 lacked appreciable pharmacological effects in each assay. Whereas pretreatment with SR141716A (3 mg kg$^{-1}$) attenuated the magnitude of cataleptic (**d**), antinociceptive (**e**) and hypothermic (**f**) effects of HU210 (1.7 mg kg$^{-1}$), HU910 did not significantly reduce HU210-induced catalepsy (**g**), antinociception (**h**) or hypothermia (**i**). Filled symbols denote significance versus the respective vehicle injection for each drug (**a–c**). ****$P < 0.0001$ HU210 versus respective pretreatment (PTx) (Holm-Sidak's test). ^^^^$P < 0.0001$ versus vehicle/HU210 (Holm-Sidak's test). Sample sizes = 8 mice/group (**a–c**) and 6 mice/group (**d–i**). All values reflect mean ± s.e.m.

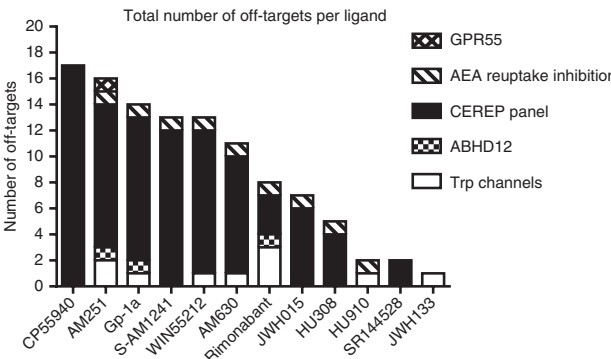

**Figure 5 | Summary of off-target activity.** Summary of the total amount of off-targets per reference ligand. CP55940 is the most nonselective one with 17 off-targets, whereas JWH133 seems to be the most selective of the available agonists. Of the antagonists, SR144528 seems to be the most selective ligand. Off-target activity in the CEREP panel, on serine hydrolases, or on AEA reuptake inhibition is defined as >50% difference in effect compared to basal levels at a concentration of 10 μM and/or a submicromolar potency on the given off-target.

potent, widely used CBR agonist CP55940 was the least selective compound at 10 μM. Consequently, it would be advisable to use it only at low concentrations (for example, <100 nM) in *in vitro* and *in vivo* experiments. Furthermore, the fact that the potent $CB_1R$ antagonists AM251 and SR141716A display $CB_2R$ antagonism at low micromolar concentrations should be taken into account in the experimental design of *in vitro*-, and possibly *in vivo* experiments. Additional off-targets of $CB_1R$ antagonists may further complicate the interpretation of results. For example, both antagonists, together with Gp-1a, target ABHD12, an enzyme that hydrolyses 2-AG (ref. 38), and may therefore increase endogenous 2-AG levels in a cell-selective manner. The low nanomolar effects of SR141716A on TRPM8 activation should also be kept in mind using cell lines expressing functional TRPM8. However, this off-target may not be functionally relevant *in vivo* because no significant effects of SR141716A on normal temperature regulation have been reported in rodent or human studies. Finally, AM251 demonstrated also some agonistic activity on GPR55 at high concentrations.

The $CB_2R$ agonists investigated in the present study displayed diverse physico-chemical properties and pharmacokinetics (p.o. or i.p. $T_{1/2}$ and oral bioavailability), which were far from optimal and deviate from standard criteria of druglikeness[39]. These factors should be taken into account during the design of *in vivo* studies. Nevertheless, our PK experiments have proven that effective drug concentrations can be achieved after intravenous as well as oral administration. Likely this can also be achieved using alternative application routes such as food admix, i.p. and s.c. However, formulations might need to be optimized for each compound and plasma concentrations need to be monitored during the course of the *in vivo* studies. The more prolonged sustained beneficial effect of HU910 in liver post-ischaemia paradigms could be explained, at least in part, by its better pharmacokinetics profile compared to HU308 and JWH133 (refs 9–11).

Lastly, HU308, HU910 and JWH133 were found to have no $CB_1$ activity *in vivo* when tested in the mouse cannabinoid triad (anti-nociception, catalepsy and hypothermia). The fourth assay of the mouse cannabinoid tetrad, hypolocomotion, was not included in this study, because the mice acclimatized to the locomotor activity chamber due to cumulative dosing procedure.

The omission of the hypolocomotor assay does not confound our conclusions, because to classify a compound as an *in vivo* active $CB_1$ receptor agonist, the ligand must be active in all four assays. Since HU308, HU910 and JWH133 did not elicit activity in the first three assays, the compounds are not active on $CB_1$ receptor *in vivo*. In conclusion, our data from the 18 compounds comprehensively and collectively studied under controlled experimental conditions indicate that HU910, HU308 and JWH133 are the most suitable $CB_2R$ agonists for preclinical target validation, based on the following observations: (a) selective agonistic activity on $CB_2R$ over $CB_1R$ in both humans and mice, (b) well-balanced activation of signal transduction pathways on $hCB_2R$, (c) minimal number of off-target activities at their active concentrations, (d) reasonable pharmacokinetics and (e) lack of functional *in vivo* pharmacological effects indicative of $CB_1R$ activity. Of the antagonists tested, SR144528 is the most suitable for use because of its high selectivity profile for $CB_2R$ in both humans and mice.

On a final note, this collaborative effort between multiple independent academic laboratories and industry to reach consensus via multicentric profiling on the key properties of widely used CBR ligands in the ever-growing field of (endo)cannabinoid research provides substantial knowledge on $CB_2$ receptor pharmacology, which may improve data reproducibility in the field. Moreover, our unique approach may serve as a useful strategy to investigate other classes of molecules with therapeutic relevance. Such an effort is deemed necessary to allow a successful transfer of preclinical data to the patient's bedside.

## Methods

**General materials.** [$^3$H]CP55940 (specific activity 141.2 Ci mmol$^{-1}$) and GF-B/GF-C filters were purchased from Perkin Elmer (Waltham, MA, USA). Anandamide [arachidonyl-5,6,8,9,11,12,14,15-$^3$H; AEA] (specific activity 200 Ci mmol$^{-1}$) was purchased from ARC (St Louis, MO, USA). Bicinchoninic acid (BCA) and BCA protein assay reagent were obtained from Pierce Chemical Company (Rochford, IL, USA). The PathHunter CHO-K1 CNR1, CNR2 and mCNR2 β-Arrestin Cell Line (catalogue numbers 93-0959C2, 93-0706C2 and 93-0472C2, respectively) and the PathHunter detection kit (catalogue number 93-0001) were obtained from DiscoveRx. Cell culture plates were purchased from Sarstedt (catalogue number 83.3902) and 384-well white walled assay plates from Perkin Elmer (catalogue number 6007680). Cannabinoid reference ligand CP55940, (Rac)-AM1241, AEA and 2-arachidonoylglycerol (2-AG) were obtained from Sigma Aldrich (St Louis, MO, USA), JWH133 and Gp-1a from Tocris Bioscience (Bristol, UK), SR144528, (S)-AM1241 and (R)-AM1241 from Cayman Chemical Company (Ann Arbor, MI, USA) and HU308, HU910, HU210, SR141716A, AM630, WIN55212-2 mesylate, and AM251 were obtained from Hoffman-La Roche (Basel, Switzerland). $\Delta^9$-THC and JWH015 were synthesized according to published procedures (see below)[40–42]. Not all compounds were tested in all labs due to legal restrictions. URB597 and JZL184 were from Cayman Chemicals (Ann Arbor, MI, USA). Activity-based probes for serine hydrolases MB064 and TAMRA-FP were synthesized according to literature[43], or were bought from Thermo Fischer Scientific (Waltham, MA, USA), respectively. All buffers and solutions were prepared using Millipore water (deionized using a MilliQ A10 Biocel, with a 0.22 μm filter) and analytical grade reagents and solvents. Buffers are prepared at room temperature (RT) and stored at 4 °C, unless stated otherwise.

**Synthetic procedures general remarks.** All reactions were performed using oven or flame-dried glassware and dry solvents. Reagents were purchased from Sigma Aldrich, Acros and Merck and used without further purification unless noted otherwise. All moisture sensitive reactions were performed under an argon atmosphere. Traces of water were removed from starting compounds by co-evaporation with toluene. $^1$H- and $^{13}$C-NMR spectra were recorded on a Bruker AV 400 MHz spectrometer at 400.2 ($^1$H) and 100.6 ($^{13}$C) MHz. Chemical shift values are reported in ppm with tetramethylsilane or solvent resonance as the internal standard (CDCl$_3$ δ 7.26 for $^1$H, δ 77.0 for $^{13}$C, CD$_3$OD: δ 3.31 for $^1$H). Data are reported as follows: chemical shifts (δ), multiplicity (s = singlet, d = doublet, dd = double doublet, td = triple doublet, t = triplet, q = quartet, quinted = quint, br = broad, m = multiplet), coupling constants $J$ (Hz), and integration. HPLC purification was performed on a preparative LC-MS system (Agilent 1200serie) with an Agilent 6130 Quadruple MS detector. High-resolution mass spectra were recorded on a Thermo Scientific LTQ Orbitrap XL. Flash chromatography was performed using SiliCycle silica gel type SiliaFlash

P60 (230–400 mesh). TLC analysis was performed on Merck silica gel 60/Kieselguhr F254, 0.25 mm.

Chemical syntheses are described in detail in the Supplementary Methods.

**Cell culture and membrane preparation.** CHOK1hCB$_1$_bgal, CHOK1hCB$_2$_bgal and CHOK1mCB$_2$_bgal cells (source: DiscoveRx, Fremont, CA, USA) were cultured in Ham's F12 Nutrient Mixture supplemented with 10% fetal calf serum, 1 mM glutamine, 50 µg ml$^{-1}$ penicillin, 50 µg ml$^{-1}$ streptomycin, 300 mg ml$^{-1}$ hygromycin and 800 µg ml$^{-1}$ geneticin in a humidified atmosphere at 37 °C and 5% CO$_2$. Cells were subcultured twice a week at a ratio of 1:20 on 10-cm ø plates by trypsinization. For membrane preparation the cells were subcultured 1:10 and transferred to large 15-cm diameter plates. For membrane preparation, the cells were detached by scraping them into 5 ml phosphate-buffered saline (PBS), collected and centrifuged at 1,000g for 5 min. Pellets derived from 30 plates were added together and resuspended in 20 ml ice-cold buffer (50 mM Tris-HCl, 5 mM MgCl$_2$, pH 7.4). An UltraThurrax homogenizer was used to homogenize the cell suspension. Membranes and the cytosolic fraction were separated by ultracentrifugation (31.000 rpm, with a Ti-70 rotor in a Beckham Coulter Ultracentrifuge) at 4 °C for 20 min. The supernatant was discarded and the pellet was resuspended in 10 ml of the same buffer and the homogenization and centrifugation steps were repeated. Supernatant was discarded and the pellet was resuspended in 5 ml buffer. Aliquots of 200 µl were frozen at −80 °C until further use. Protein concentration was determined using the BCA method[44].

**[3H]CP55940 displacement assay.** [3H]CP55940 displacement assays were used for the determination of affinity (K$_i$) values of ligands for the cannabinoid CB$_1$ and CB$_2$ receptors. Membrane aliquots containing 5 µg (CHOK1hCB$_1$_bgal) or 1 µg (CHOK1hCB$_2$_bgal) of membrane protein in 100 µl assay buffer (50 mM Tris-HCl, 5 mM MgCl$_2$, 0.1% bovine serum albumin (BSA), pH 7.4) were incubated at 30 °C for 1 h, in presence of 3.5 nM [3H]CP55940 (CHOK1hCB$_1$_bgal) or 1.5 nM [3H]CP55940 (CHOK1hCB$_2$_bgal). Non-specific binding was determined in the presence of 10 µM AM630 (CHOK1hCB$_2$_bgal) or 10 µM SR141716A (CHOK1hCB$_1$_bgal). When AEA or 2-AG were used as ligands, phenylmethylsulphonyl fluoride was used as a serine protease inhibitor. Incubation was terminated by rapid filtration performed on GF/C filters (Whatman International, Maidstone, UK), presoaked for 30 min with 0.25% PEI, using a Brandel harvester (Brandel, Gaithersburg, MD, USA). Filter-bound radioactivity was determined by scintillation spectrometry using a Tri-Carb 2900 TR liquid scintillation counter (Perkin Elmer, Boston, MA, USA). For mouse experiments, brain (for CB$_1$) and spleen (for CB$_2$) were resuspended in 2 mM Tris-EDTA, 320 mM sucrose, 5 mM MgCl$_2$ (pH 7.4), then homogenized in a Potter homogenizer and centrifuged three times at 1000g (10 min). The supernatants were centrifuged at 18000g (30 min), and the pellets were resuspended in assay buffer (50 mMTris-HCl, 2 mMTris-EDTA, 3 mM MgCl$_2$, pH 7.4). These membrane fractions were used in rapid filtration assays with 400 pM of [3H]CP55,940. In all binding experiments, nonspecific binding was determined in the presence of 1 µM 'cold' agonist. When AEA or 2-AG were used as ligands, the FAAH inhibitor URB597 or the MAGL inhibitor JZL184, respectively, were included in the assay buffer[45]. Brain and spleen tissue was collected from male C57BL/6J mice, 8 months old (supplier: Charles River Via Indipendenza 11 Calco (Lecco) 23885 Italy), at the Santa Lucia Foundation within the project: 'Ruolo del Sistema Endocannabinoide nei Processi Neurodegenerativi in modelli animali di Alzheime' by Mauro Maccarrone. Ethical approval was given by the Ministry of Health, n. 47/2014/PR (deadline 17 November 2019).

**[35S]GTPγS assay.** G-protein activation by the receptor is measured by the binding of radiolabelled GTP, [35S]GTPγS, to the receptor[46]. Five micrograms of homogenized CHOK1CB_bgal membranes in 20 µl assay buffer (50 mM Tris-HCl buffer (pH 7.4), 5 mM MgCl$_2$, 150 mM NaCl, 1 mM EDTA, 0.05% BSA and 1 mM DTT, freshly prepared every day) were pretreated with 5 µg saponin and 1 µM GDP. To determine the pEC$_{50}$ and Emax values of cannabinoid reference ligands, the membranes were incubated with various concentrations of the ligands of interest. The basal level of GTP binding was measured in untreated membrane samples, and the maximal level of GTP binding was measured by treatment of the membranes with 10 µM CP55940. All samples were preincubated for 30 min at rt, followed by addition of [35S]GTPγS (0.3 nM). The total value of each sample was 100 µl. The samples were incubated for 90 min at 25 °C on a shaking platform. Incubations were terminated by rapid vacuum filtration to separate the bound and free radioligand through Whatman GF/B filters (Perkin Elmer, Groningen, The Netherlands) using a Filtermate 96-well harvester (Perkin Elmer, Groningen, The Netherlands). Filters were subsequently washed ten times with ice-cold wash buffer containing 50 mM Tris HCl, pH 7.4 and 5 mM MgCl$_2$. Microscint scintillation fluid (25 µl, Perkin Elmer, Groningen, The Netherlands) was added to each filter. After 3 h, the filter-bound radioactivity was determined by scintillation spectrometry using a 1450 Microbeta Wallac Trilux counter (Perkin Elmer).

**PathHunter β-arrestin recruitment assay.** The assay was performed using the PathHunter hCB$_1$_bgal, hCB$_2$_bgal, hGPR55_bgal or mCB$_2$_bgal CHOK1 β-arrestin recruitment assay kit (source: DiscoveRx, Fremont, CA, USA), according

to the manufacturer's protocol[47]. Briefly, cells were seeded at a density of 5,000 cells per well of solid white walled 384-well plates (Perkin Elmer, MA, USA) in 20 µl cell culture medium and incubated overnight (16–18 h) in a humidified atmosphere at 37 °C and 5% CO$_2$. In the endocannabinoid assay, a concentration of 50 µM of phenylmethylsulphonyl fluoride was freshly added to each well and the cells were incubated for 30 min in a humidified atmosphere at 37 °C and 5% CO$_2$. In the agonistic assay, the cells were stimulated with 10 µM of each agonist (single point assay) or 11 increasing concentrations of each agonist and incubated for 90 min in a humidified atmosphere at 37 °C and 5% CO$_2$. In the antagonistic assays, the cells were exposed to 10 µM of each antagonist (single point assay) or 11 increasing concentrations of each antagonist and incubated for 30 min in a humidified atmosphere at 37 °C and 5% CO$_2$, followed by the addition of the EC80 concentration of CP55940 (25 nM for CHOK1hCB1_bgal and 46 nM for CHOK1hCB2_bgal). The cells were incubated for 90 min in a humidified atmosphere at 37 °C and 5% CO$_2$. Compounds in DMSO stocksolutions were added using a HP D300 Digital Dispenser (Tecan, Männedorf, Switzerland). Endocannabinoids 2-AG and anandamide (stocksolutions in acetonitrile) were added manually. The final concentration of organic solvent per assay point was ≤0.1%. β-Galactosidase enzyme activity was determined by using the PathHunter Detection mixture, according to the kit's protocol[47]. Detection mixture, 12 µl per well, was added and the plate was incubated for 1 h in the dark at room temperature. Chemiluminescence, indicated as relative light unit, was measured on an EnVision multilabel plate reader (Perkin Elmer, MA, USA).

**pERK assay.** Assays to determine pERK stimulation were performed in HEK cells stably expressing 3HA-tagged human CB$_1$ receptors and Flp-in HEK cells stably expressing 1HA-3TCS-tagged hCB$_2$ receptors (3TCS refers to three 'thrombin cleavage sites', which are not relevant for these experiments). HEK-hCB$_1$ cells were created by transfecting Human Embryonic Kidney 293 (HEK) cells (ATCC #CRL-1573) with human CB$_1$ (hCB$_1$) chimerized with three hemeagglutinin (HA) tags at the amino terminus. A single HA tag, three thrombin cleavage sites and a single 'L' residue (for cloning purposes) were chimerized at the N terminus of hCB$_2$ (Missouri S&T cDNA Resource Center, www.cdna.org, CNR0200000) by overlap-extension PCR and restriction digest/ligation from DNA oligonucleotides. The subsequent hCB2 construct was cloned into pcDNA5/FRT (Life Technologies #V601020) via the NheI and XhoI sites using standard molecular biology techniques and sequence verified. The human Flp-In-293 cell line (Invitrogen #R750-07) was stably transfected with the human hCB2 construct for targeted integration into the FRT site and maintained in 50 µg hygromycin B ml$^{-1}$ (ref. 48). Cells were grown in high glucose DMEM, supplemented with 10% fetal bovine serum (FBS) and were cultured under standard conditions. Briefly, cells from semi-confluent T75 flasks were seeded in 100 µl per well complete medium, in 96-well plates pre-treated with poly-D-lysine. Approximately 24 h after seeding, cells were 60% confluent and complete medium was removed and replaced with 50 µl per well serum-free DMEM containing 1 mg ml$^{-1}$ BSA, for 18 h to reduce the pERK background. Drugs were made up in DMEM + 1 mg ml$^{-1}$ BSA at 2× concentrations and cells were stimulated by dispensing 50 µl per well in a 37 °C waterbath for exactly 5 min. At the conclusion of stimulation, plates were moved to an ice bed and well contents were rapidly aspirated. Thirty microliters per well of lysis buffer was immediately dispensed, and plates were placed on shakers for 10 min prior to proceeding to detection. Lysis buffer and detection reagents were from a PerkinElmer AlphaScreen SureFire pERK (Thr202/Tyr204) assay kit, and were utilized according to the manufacturer's instructions. In order to reduce use of detection reagents, lysates were first transferred to 1/2 area, white 96-well plates (PerkinElmer). Detection of hCB$_1$-mediated pERK responses was performed using the manufacturer's standard detection protocol, whereas hCB$_2$ pERK responses were only detected utilizing the high sensitivity detection protocol. Plates were read on an EnSpire plate reader (PerkinElmer).

**GIRK assay.** Activation of native GIRK channels in mouse AtT20 neuroblastoma cells stably transfected with HA-tagged hCB$_1$ or hCB$_2$ receptors (source: cells were generated by transfecting WT AtT20 with HA-tagged hCB$_1$ or hCB$_2$ receptor. The cDNA clones for the human CB$_1$ and CB$_2$ receptor with 3×N-terminal HA tags were obtained from the Missouri S&T cDNA Resource Center (www.cdna.org)) was measured as a change in membrane potential using membrane potential-sensitive dye (blue, Molecular Devices) in a FlexStation three-plate reader[49,50]. Briefly, AtT20 cells from an 80–90% confluent 75 mm$^2$ flask were resuspended in L-15 medium supplemented with 1% FBS, 100 U penicillin and 100 µg streptomycin ml$^{-1}$ and plated in 96-well black-walled plates (90 µl per well). Cells were incubated overnight in humidified room air at 37 °C. Membrane potential dye was dissolved in low K HEPES buffered saline (90 µl) and added to the plate an hour before assay. Fluorescence was measured every 2 s (λexcitation = 530 nm, λemission = 565 nm). Assays were carried out at 37 °C, drugs were added in volumes of 20 µl after at least 2 min of baseline recording. Changes in fluorescence were expressed as a percentage of predrug values, after subtraction of the small changes produced by solvent (0.1% DMSO, 0.1% BSA) alone. For antagonist experiments, drugs were preincubated for at least 5 min before CP55940 (30 nM) was added. For each drug concentration, the peak change in fluorescence was normalized to the change produced a maximally effective

concentration of CP55–940 (1 µM). Data are expressed as the mean ± s.e.m. of *N* independent experiments, each performed in duplicate.

**Mouse cAMP assay.** The effect of ligands on the forskolin-stimulated accumulation of cAMP was determined with the LANCE ULTRA cAMP kit (Perkin-Elmer Life Sciences, Boston, MA, USA) (ref. 51). Brain (for CB₁) and spleen (for CB₂) were resuspended in 50 mM of Tris-HCl (pH 7.4), then were homogenized in a Potter homogenizer and centrifuged at 1000g for 10 min. The supernatants were incubated for 30 min with 1-methyl-3-isobutylxanthine (IBMX), then forskolin was added in the presence or absence of ligands. The supernatants were incubated for 30 min in ATP Regeneration Buffer (50 mM HEPES, pH 7.4, 10 mM phosphocreatine, 10 units ml⁻¹ creatine phosphokinase, 10 µM GTP, 200 µM ATP, 10 mM MgCl₂, 250 µM IBMX), and the reaction was stopped by adding lysis buffer. Time-resolved fluorescence was measured with a Victor V Multilabel counter (Perkin-Elmer Life Sciences, Boston, MA, USA). Brain and spleen tissue was collected from male C57BL/6J mice, 8 months old (supplier: Charles River Via Indipendenza 11 Calco (Lecco) 23885 Italy), at the Santa Lucia Foundation within the project: 'Ruolo del Sistema Endocannabinoide nei Processi Neurodegenerativi in modelli animali di Alzheime' by Mauro Maccarrone. Ethical approval was given by the Ministry of Health, n. 47/2014/PR (deadline 17 November 2019).

**Human cAMP assay.** cAMP assays were performed with CHO cells stably expressing human CB₁ or human CB₂ receptors[52] (source: DiscoveRx, Fremont, CA, USA) using the cAMP-Nano-TRF detection kit (Roche Diagnostics, Penzberg, Germany). Cells were seeded 17–24 h prior to the experiment at a density of 3 × 10⁴ cells per well in a black 96-well plate with flat clearbottom (Corning, Wiesbaden, Germany) and incubated in 5% CO₂ at 37 °C in a humidified incubator. The growth medium was exchanged with Krebs Ringer bicarbonate buffer with 1 mmol l⁻¹ 3-isobutyl-1-methylxanthine (IBMX), 0.1% fatty acid-free BSA and incubated at 30 °C for 60 min. Agonist was added to a final assay volume of 100 µl and the mixture was incubated for 30 min at 30 °C. The assay was stopped by the addition of 50 µl 3 × lysis reagent and shaken for 2 h at room temperature. The time-resolved energy transfer was measured using an LF502 Nanoscan FLT (IOM, Berlin, Germany), equipped with a laser as excitation source. cAMP content was determined from the function of a standard curve spanning from 10 to 0.13 nmol l⁻¹ cAMP.

**Data analysis of functional assays.** All experimental data were analysed using the nonlinear regression curve fitting program GraphPad Prism 6.0 (GraphPad Software, Inc., San Diego, CA, USA). From displacement assays, pIC₅₀ values were obtained by non-linear regression analysis of the displacement curves. The obtained pIC₅₀ values were converted into pK₁ values using the Cheng Prusoff equation to determine the affinity of the ligands (K_D: 0.33 (CB₂R), 0.10 (CB₁R)) (ref. 53). β-Arrestin recruitment and GTPγS curves were analysed by the nonlinear regression option 'log (agonist or inhibitor) versus response-variable slope' to obtain potency, inhibitory potency or efficacy values of agonists and inverse agonists (EC₅₀, IC₅₀ or E_max, respectively). Basal activity of the cells is set at 0%. For the β-arrestin recruitment assay, all data points were corrected for any background (for example, background luminescence). For the analysis of antagonists measured in competition with CP55940, the nonlinear regression option 'log (inhibitor) versus response' was chosen. The response of agonists per sample is normalized to the effect of 10 µM CP55940 and the response of antagonists is normalized to effect of the EC₈₀ of CP55940. For the GTPγS assay, agonistic effect is normalized to the effect of 10 µM CP55940. Data shown are the mean ± s.e.m. of at least three independent experiments, each performed in duplicate, unless stated otherwise.

**Data analysis for bias calculations.** The data used for the operation analysis was the data of at least three independent experiments on each assay, all normalized to the effect of 10 µM CP55940. The analysis used was based on van der Westhuizen *et al.*[30] For more details about the discovery of the operational model and its mathematical background, see Black and Leff and Kenakin *et al.*[31,54,55] See Supplementary Methods for a full step-by-step procedure.

**AEA reuptake inhibition in HaCaT cells.** The uptake of [³H]AEA was measured in intact HaCaT cells (a kind gift of Prof N.E. Fusenig (German Cancer Research Center, Heidelberg, Germany), that were incubated in PBS at 37 °C with a mixture of AEA [arachidonyl-5,6,8,9,11,12,14,15-³H] (200 Ci mmol⁻¹) and cold AEA (at a final concentration of 400 nM) for 15 min (ref. 56). Control experiments were carried out also at 4 °C and in the presence of the selective AEA reuptake inhibitor OMDM-1 (10 µM). The effect of different ligands on AEA reuptake was tested by adding each substance directly to the incubation medium.

**AEA reuptake inhibition in HMC-1 cells and U937 cells.** Compound screening and IC50 determinations for AEA cellular uptake in U937 cells and HMC-1 cells (source: U937 cells were purchased from ATCC, Manassas, VA, USA, HMC-1

cells were a gift of Prof S. Ständer, University of Münster with the permission of the Mayo Foundation, USA) was performed in 96-well format using AquaSil silanized glass vials (Chromacol 1.1-MTV) (refs 32,57). First, required amounts of U937 cells were centrifuged at 100g for 5 min and resuspended in RPMI (37 °C) to a final concentration of 2 × 10⁶ cells ml⁻¹. Then, 250 µl of cell suspension (0.5 × 10⁶ cells per sample) were transferred into the glass vials. After addition of 5 µl vehicle (DMSO) or test compounds dissolved in DMSO at indicated final concentrations the cells were incubated at 37 °C for 15 min. After pre-incubation a mixture of 0.5 nM [ethanolamine-1-³H]-AEA, (60 Ci mmol⁻¹) and 99.5 nM of cold AEA (final 100 nM) was added and the samples were incubated at 37 °C for another 15 min. In the FAAH-deficient HMC-1 cells, [arachidonoyl-5,6,8,9,11,12,14,15-³H]-AEA (200 Ci mmol⁻¹) was used in order to achieve a better signal-to-background ratio, due to the lower rate of AEA uptake. The reaction was stopped by rapid filtration over UniFilter-96 GF/C filters (Perkin Elmer) pre-soaked with PBS 1% BSA. Cells on filters were washed three times with 100 µl ice-cold PBS buffer containing 1% fatty acid free BSA. After drying, 45 µl MicroScint 20 scintillation cocktail (Perkin Elmer, Waltham, MA, USA) was added to the wells and the plate was sealed. Radioactivity was measured by liquid scintillation counting on a Perkin Elmer Wallac Trilux MicroBeta 1450 during 2 min. Samples were measured in triplicates in *n* = 3 independent experiments except the screening run (*n* = 1). Each run was validated using the positive controls OMDM-2 and UCM707 at a concentration of 10 µM reaching 59.7 ± 6.6% (*n* = 7) and 71.5 ± 8.0% (*n* = 7) respectively.

**NAPE-PLD inhibition.** Full length human cDNA NAPE-PLD was obtained from Natsuo Ueda[58] and cloned into mammalian expression vector pcDNA3.1, containing a C-terminal Flag-tag and genes for ampicillin and neomycin resistance. All plasmids were grown in XL-10 Z-competent cells and prepped (Maxi Prep, Qiagen). Sequence analysis for the confirmation of the sequences was performed at the Leiden Genome Technology Centre.
    HEK293T cells (source: Dutch Cancer Institute) were cultured at 37 °C and 7% CO₂ in DMEM with glutamax, penicillin (100 µg ml⁻¹), streptomycin (100 µg ml⁻¹) and 10% New Born Calf Serum iron supplemented (Hyclone SH30072.03). Cells were passaged twice a week to appropriate confluence. Twenty-four hours before transfection 10⁷ cells were seeded on a 15 cm dish. Two hours before transfection, the medium was refreshed. Transfection is performed with PEI in a ratio of 3:1 with human NAPE-PLD or Mock pcDNA3.1Neo, 20 µg per dish. Medium is refreshed after 24 h and cells are harvested after 72 h. Cell suspensions are centrifuged at 1,000g for 10 min, supernatant removed and pellets frozen at − 80 °C until further use.
    Cell pellets are re-suspended in lysis buffer 1: 20 mM Hepes, 2 mM DTT, 0.25 M sucrose, 1 mM MgCl₂, 2.5 U ml⁻¹ benzonase and incubated 30 min on ice. The cytosolic fraction (supernatant) is separated from the membranes by ultracentrifugation (32,000 rpm for 30 min 100,000g). The pellet is resuspended in buffer 2: 20 mM Hepes, 2 mM DTT (membrane fraction). All samples are stored at − 80 °C. Enzyme concentrations are determined using a Bradford assay[59].
    The membrane protein fraction from transient overexpression of NAPE-PLD in HEK293T cells was diluted to 0.4 mg ml⁻¹ in assay buffer: 50 mM Tris-HCl (pH 7.5), 0.02% Triton X-100, 150 mM NaCl (ref. 60). The substrate PED6 (Invitrogen) 10 mM stock was consecutively diluted in DMSO (25 ×) and in assay buffer (10 ×). Relevant concentrations of compounds are prepared in DMSO. The assay is performed in a dark Greiner 96-well plate, end volume 100 µl. The compound or DMSO is incubated with membrane protein lysate (final concentration 0.04 mg ml⁻¹) for 30 min at 37 °C. A sample without membrane protein lysate is incorporated for background subtraction. Then, substrate is added (final concentration 1 µM) and the measurement is started immediately on a TECAN infinite M1000 pro at 37 °C (excitation 485 nm, emission 535 nm), scanning every 2 min for 1 h.

**Cell culture and membrane preparation for DAGL, MAGL assay.** Cell culture and membrane preparation were performed as previously described[43]. HEK293T cells (source: Dutch Cancer Institute) were grown in DMEM with stable glutamine and phenolred (PAA), 10% New Born Calf serum, penicillin and streptomycin. Cells were passaged every 2–3 days by resuspending in medium and seeding them to appropriate confluence. Membranes were prepared from transiently transfected HEK293T cells. One day prior to transfection 10⁷ cells were seeded in a 15 cm petri dish. Cells were transfected with a 3:1 mixture of polyethyleneimine (60 µg) and plasmid DNA (20 µg) in 2 ml serum-free medium. The medium was refreshed after 24 h, and after 72 h the cells were harvested by suspending them in 20 ml medium. The suspension was centrifuged for 10 min at 1000 rpm, and the supernatant was removed. The cell pellet was stored at − 80 °C until use. Cell pellets were thawed on ice and suspended in lysis buffer A (20 mM HEPES, 2 mM DTT, 0.25 M sucrose, 1 mM MgCl₂, 25 U µl⁻¹ Benzonase). The suspension was homogenized by polytrone (3 × 7 s) and incubated for 30 min on ice. The suspension was subjected to ultracentrifugation (100,000 × g, 30 min, 4 °C, Beckman Coulter, Type Ti70 rotor) to yield the cytosolic fraction in the supernatant and the membrane fraction as a pellet. The pellet was resuspended in lysis buffer B (20 mM HEPES, 2 mM DTT). The protein concentration was determined with Quick Start Bradford assay (Biorad). The protein fractions were

diluted to a total protein concentration of 1 mg ml$^{-1}$ and stored in small aliquots at $-80\,°C$ until use.

**Biochemical DAGLα activity assay.** hDAGL-α activity was measured by the extent of the hydrolysis of para-nitrophenylbutyrate (PNP-butyrate) by membrane preparations from HEK293T cells (source: Dutch Cancer Institute) transiently transfected with hDAGL-α (ref. 43). Percentage of inhibition of reference ligands was determined in comparison with an untreated control. Reactions were performed in 50 mM pH 7.2 HEPES buffer with 0.05 µg µl$^{-1}$ final protein concentration hDAGL-α transfected protein.

**Biochemical MAGL activity assay.** MAGL activity was measured by the extent of the production of glycerol from 2-AG hydrolysis by MAGL-overexpressing membrane preparations from transiently transfected HEK293T cells (source: Dutch Cancer Institute)[61]. Percentage of inhibition of reference ligands was determined in comparison with an untreated control. Standard assay conditions are as follows: 0.2 U ml$^{-1}$ glycerol kinase (GK), glycerol-3-phosphate oxidase (GPO) and horseradish peroxidase (HRP), 0.125 mM ATP, 10 µM AmplifuRed, 5% DMSO, 25 µM 2-AG and 0.5% acetonitrile in a total volume of 200 µl. Fluorescence was measured in 5 min intervals for 60 min.

**FAAH activity.** FAAH activity was assessed using U937 cell homogenate (source: U937 cells were purchased from ATCC, Manassas, VA, USA). Tested compounds (at the concentrations of 1 and 5 µM) or solvent ($\leq 1\%$ of the final volume) were pre-incubated at 37 °C for 30 min with 100 µg of cell homogenate diluted in assay buffer (10 mM Tris-HCl and 1 mM EDTA, pH 7.6 and 1% w/v fatty acid-free BSA). Successively, a mixture of 99.5 nM nM of AEA and 0.5 nM of [ethanolamine-1-3H]-AEA (40–60 Ci mmol$^{-1}$) was added to the homogenate (final concentration of 100 nM) and incubated for 10 min at 37 °C under shaking. After the incubation time, 2 volumes of an ice-cold methanol:chloroform mixture 1:1 (v/v) were added to each sample, vigorously vortexed and centrifuged at 10,000$g$ for 10 min at 4 °C to separate aqueous and organic phases. The aqueous phases were collected and the radioactivity associated with the hydrolysis product [3H]-ethanolamine was measured after addition of 3 ml of scintillation cocktail using the Tri-Carb 2100 TR scintillation counter. Data were collected from three independent experiments performed in triplicate and results were expressed as FAAH activity, relative to that in vehicle-treated samples ($=100\%$).

**COX2 activity.** The COX2 activity was assessed using a COX fluorescent inhibitor screening assay kit from Cayman chemicals. The assay was performed in a final volume of 50 µl in black 384-well non-binding microplates. Tested compounds (at the screening concentration of 5 µM, 2.5 µl) or solvent were incubated for 15 min at 37 °C with the following solution: 37.5 µl of assay buffer (Tris-HCl, 100 mM, pH = 8), 2.5 µl of heme (final concentration of 1 µM), 2.5 µl of COX-2 enzyme and 2.5 µl of 10-acetyl-3,7-dihydroxyphenoxazine (ADHP) fluorometric substrate (final concentration of 30 µM). The reaction was started by addition of arachidonic acid (AA) or 2-AG (2.5 µl, final concentration of 10 µM). Fluorescence signals were measured after 5 min of incubation with a TECAN FARCyte Ultra (Ex. 535 nm, Em. 580 nm). The COX inhibitor DuP-697 (1 µM) was used as a positive control. Spontaneous non-enzymatic cleavage of ADHP was quantified in samples where COX2 was replaced by assay buffer. The non-specific value was then subtracted from the measured fluorescence. Data were collected from three independent experiments performed in triplicate and results were expressed as COX2 activity, relative to that in vehicle-treated samples ($=100\%$).

**ABHD6 and ABHD12 activity.** HEK293T cells were obtained from American Type Culture Collection (ATCC, Manassas, VA, USA). Lipofectamine 2000 was obtained from Life Technologies (Carlsbad, CA, USA) and geneticin from InvivoGen. Cell culture media, FBS, and cell culture supplements were from Invitrogen (Carlsbad, CA, USA). pCMV6-AC-hABHD6 and pCMV6-XL4-hABHD12 were a kind gift from Prof Jarmo T. Laitinen. hABHD12 sequence was subsequently inserted in a pcDNA3.1 plasmid in order to generate a stable cell line. The effect of CB$_2$R ligands on ABHD6 and ABHD12 hydrolytic activity was determined using cell homogenate from hABHD6 and hABHD12 overexpressing HEK293T cells (40 µg total protein per condition). Samples were pre-incubated with the compounds (10 µM in the screening assay and concentrations in the range 1 nM—50 µM for the dose-response curves) in Tris 1 mM, EDTA 10 mM (pH = 7.6) buffer containing 0.1%(w/v) BSA. DMSO was used as vehicle control and WWL70 10 µM or THL 20 µM as positive controls. [3H]-2-OG (10 µM final concentration) was added and after incubation of 5 min at 37 °C, the reaction was stopped by the addition of 400 µl of ice-cold CHCl$_3$:MeOH (1:1). The samples were vortex and centrifuged (16,100$g$ 10 min 4 °C). Aliquots (200 µl) of the aqueous phase were assayed for tritium content by liquid scintillation spectroscopy. The values obtained for each measurement were corrected for non-specific hydrolysis (non-ABHD6 mediated) by subtracting the signal obtained in the hydrolysis of [3H]- 2-OG by non-transfected HEK293T.

**Activity-based protein profiling in mouse brain tissue.** Mouse tissue was isolated according to guidelines approved by the ethical committee of Leiden University (DEC#13191) (ref. 43). Animal care, tissue isolation and the following experimental procedures was all performed at the lab of Mario van der Stelt, Leiden University. Mouse tissues were homogenized in pH 7.2 lysis buffer (20 mM Hepes, 2 mM DTT, 1 mM MgCl$_2$, 25 U ml$^{-1}$ Benzonase) and incubated for 5 min on ice, followed by low speed spin (2,500$g$, 3 min, 4 °C) to remove debris. The suspension was subjected to ultracentrifugation (100,000$g$, 45 min, 4 °C, Beckman Coulter, Type Ti70 rotor) to separate the cytosolic fraction from the membrane fraction. The pellet was resuspended in storage buffer (20 mM Hepes, 2 mM DTT). The total protein concentration was determined with Quick Start Bradford assay (Biorad) or Qubit protein assay (Invitrogen). Tissue proteome (2.5 mg ml$^{-1}$) was incubated with vehicle (DMSO) or inhibitor in 0.5 µl DMSO (10 µM final concentration) for 30 min at rt, and subsequently incubated with broad spectrum ABPs MB064 (250 nM final concentration) or TAMRA-FP (500 nM final concentration) for 15 min at rt (ref. 43). The reactions were quenched with 10 µl standard 3 × SDS-PAGE sample buffer. The samples were directly loaded (7 µl = ~12 µg) and resolved on SDS page gel (10% acrylamide). The gels were visualized and scanned with a Bio-Rad Universal Hood III (Bio-Rad Laboratories B.V.) using Cy3/TAMRA settings (excitation wavelength 532 nm, emission wavelength 580 nm).

**Off-target activity on TRP channels.** HEK293 (human embryonic kidney) cells stably over-expressing recombinant human TRPV1 or rat TRPA1, TRPV2, TRPV3, TRPV4 or TRPM8 (source: human embryonic kidney cells were purchased by DSMZ (Germany), TRPV1-HEK-293 cells were a kind gift from John Davis, GlaxoSmithKline, Harlow, UK, the plasmid for TRPV2, as well as TRPV3-HEK-293 and TRPV4-HEK-293 were a kind gift from HB Bradshaw Indiana University, the plasmid containing TRPA1 was a kind gift from Sven-Eric Jordt then at Department of Cellular and Molecular Pharmacology University of California, San Francisco, California, USA and now at Department of Anesthesiology, Duke University School of Medicine, Durham, NC, USA, TRPM8-HEK-293 was a gift from Mario van der Stelt) were grown on 100 mm diameter Petri dishes as mono-layers in minimum essential medium supplemented with non-essential amino acids, 10% FBS, 2 mM glutamine, and maintained at 5% CO$_2$ at 37 °C. Quantitative real-time analysis was carried out to measure TRP gene over-expression in transfected-cells (data not shown). On the day of the experiment, cells were loaded with the methyl ester Fluo-4 AM in minimum essential medium (4 µM in DMSO containing 0.02% Pluronic F-127, Invitrogen), kept in the dark at room temperature for 1 h, washed twice with Tyrode's buffer (145 mM NaCl, 2.5 mM KCl, 1.5 mM CaCl$_2$, 1.2 mM MgCl$_2$, 10 mM D-glucose, and 10 mM HEPES, pH 7.4), resuspended in the same buffer and transferred (about 100,000 cells) to the quartz cuvette of the spectrofluorimeter (Perkin-Elmer LS50B equipped with PTP-1 Fluorescence Peltier System; PerkinElmer Life and Analytical Sciences, Waltham, MA, USA) under continuous stirring. The effects on intracellular Ca$^{2+}$ concentration ([Ca$^{2+}$]$_i$) before and after the addition of various concentrations of test compounds was measured by cell fluorescence ($\lambda_{EX} = 488$ nm, $\lambda_{EM} = 516$ nm) at 25 °C. The effects of compounds were normalized against the response to ionomycin (4 µM) in each experiment. The increases in fluorescence in wild-type HEK293 cells (that is, not transfected with any construct) were used as baseline and subtracted from the values obtained from transfected cells. Efficacy was defined as the maximum response elicited by the compounds tested and was determined by comparing their effect with the analogous effect observed with 4 µM ionomycin (Cayman), while the potency of the compounds (EC$_{50}$) was determined as the concentration required to produce half-maximal increases in [Ca$^{2+}$]$_i$. Curve fitting (sigmoidal dose-response variable slope) and parameter estimation were performed with GraphPad Prism (GraphPad Software Inc., San Diego, CA, USA).

Antagonist/desensitizing behaviour was evaluated by adding the test compounds in the quartz cuvette 5 min before stimulation of cells with agonists. In the case of human TRPV1-expressing HEK293 cells the agonist used was capsaicin (0.1 µM, in the case of SR141716A 10 nM was also used), which was able of elevating intracellular Ca$^{2+}$ with a potency of EC$_{50} = 5.3 \pm 0.4$ nM and efficacy = 78.6 ± 0.6%.

For TRPV2, the rat TRPV2-HEK293 cells exhibited a sharp increase in [Ca$^{2+}$]$_i$ upon application of lysophosphatidylcholine (LPC) 3 µM. The concentration for half-maximal activation was 3.40 ± 0.02 µM and efficacy was 91.7 ± 0.5%.

In the case of TRPV3, rat TRPV3-expressing HEK-293 cells were first sensitized with the non-selective agonist 2-aminoethoxydiphenyl borate (100 µM). Antagonist/desensitizing behaviour was evaluated against thymol (100 µM), which showed an efficacy of 34.7 ± 0.2% and a potency of EC$_{50} = 84.1 \pm 1.6$ µM.

In the case of rat TRPV4-expressing HEK-293 cells the agonist used was 4α-phorbol 12,13-didecanoate (4α-PDD) (1 µM), which was able of elevating intracellular Ca$^{2+}$ with a potency of EC$_{50} = 0.46 \pm 0.07$ µM, and an efficacy of 51.9 ± 1.7%.

In the case of rat TRPM8-expressing HEK-293 cells, antagonist/desensitizing behaviour was evaluated against icilin at 0.25 µM and 0.10 µM. For icilin, efficacy was 75.1 ± 1.1 and potency EC$_{50} = 0.11 \pm 0.01$ µM.

In the case of HEK-293 cells stably over-expressing recombinant rat TRPA1, the effects of TRPA1 agonists are expressed as a percentage of the effect obtained with

100 µM allyl isothiocyanate, which showed a potency of $EC_{50} = 1.41 \pm 0.04$ µM and an efficacy of $65.9 \pm 0.5$.

The effect on $[Ca^{2+}]_i$ exerted by agonist alone was taken as 100%. Data are expressed as the concentration exerting a half-maximal inhibition of agonist-induced $[Ca^{2+}]_i$ elevation ($IC_{50}$), which was calculated again using GraphPad. All determinations were performed at least in triplicate. Statistical analysis of the data was performed by analysis of variance at each point using ANOVA followed by the Bonferroni's test.

**CEREP panel.** Pharmacological profiles of the test compounds were generated. Data shown give the percentage of inhibition for binding assays and the percentage of inhibition for enzyme and cell-based assays at a test concentration of 10 µM. The assays were performed at CEREP according to standard procedures described under http://www.cerep.fr/cerep/users/index.asp. For the first compounds being evaluated (WIN55212-2, CP55940, SR141716A, JWH133, HU308, Gp-1a and HU910) standard profiles as suggested by CEREP were applied. For the other compounds reduced panels concentrating on most predictive antitargets and omitting rarely hit sites were used.

**Microsomal clearance.** For human or mice, pooled commercially available microsome preparations from liver tissues are used (source: pooled human microsomes (BD UltraPool HLM 150,Lot 38289) and pooled male mouse microsomes (C57BL/6J, Lot 4339006) were purchased from Corning Incorporated (Woburn, USA))[62]. For human, ultra-pooled (150 mixed gender donors) liver microsomes are purchased to account for the biological variance in vivo. For the microsome incubations, 96 deep well plates are applied, which are incubated at 37 °C on a TECAN (Tecan Group Ltd, Switzerland) equipped with Te-Shake shakers and a warming device (Tecan Group Ltd, Switzerland). The incubation buffer is 0.1 M phosphate buffer at pH 7.4. The NADPH regenerating system consists of 30 mM glucose-6-phosphate disodium salt hydrate; 10 mM NADP; 30 mM $MgCl_2 \times 6\ H_2O$ and 5 mg ml$^{-1}$ glucose-6-phosphate dehydrogenase (Roche Diagnostics) in 0.1 M potassium phosphate buffer pH 7.4.

Incubations of a test compound at 1 µM in microsome incubations of 0.5 mg ml$^{-1}$ plus cofactor NADPH are performed in 96-well plates at 37 °C. After 1, 3, 6, 9, 15, 25, 35 and 45 min 40 µl incubation solutions are transferred and quenched with 3:1 (v/v) acetonitrile containing internal standards. Samples are then cooled and centrifuged before analysis by LC-MS/MS.

Log peak area ratios (test compound peak area/internal standard peak area) are plotted against incubation time using a linear fit. The calculated slope is used to determine the intrinsic clearance: Clint (µl min$^{-1}$ per mg protein) $= -$ slope (min $- 1) \times 1,000/$[protein concentration (mg ml)$^{-1}$]. Data are obtained from single experiments measured with multiple time-points.

**Hepatocyte clearance.** For animals, hepatocyte suspension cultures are either freshly prepared by liver perfusion studies or prepared from cryopreserved hepatocyte batches[63]. For human, commercially available, pooled (5–20 donors), cryopreserved human hepatocytes from non-transplantable liver tissues are mainly used (source: primary, pooled human cryopreserved hepatocytes (Lot ECO) from nontransplantable liver tissues and pooled C57BL6 mouse hepatocytes (Lot PJJ) were purchased from BioreclamationIVT (NY, USA)). For the suspension cultures, Nunc U96 PP-0.5 ml (Nunc Natural, 267245) plates are used, which are incubated in a Thermo Forma incubator from Fischer Scientific (Wohlen, Switzerland) equipped with shakers from Variomag Teleshake shakers (Sterico, Wangen, Switzerland) for maintaining cell dispersion. The cell culture medium is William's media supplemented with Glutamine, antibiotics, insulin, dexamethasone and 10% FCS.

Incubations of a test compound at 1 µM test concentration in suspension cultures of 1 Mio cells ml$^{-1}$ ($\sim$1 mg ml$^{-1}$ protein concentration) are performed in 96-well plates and shaked at 900 rpm for up to 2 h in a 5% $CO_2$ atmosphere and 37 °C. After 3, 6, 10, 20, 40, 60 and 120 min 100 µl cell suspension in each well is quenched with 200 µl methanol containing an internal standard. Samples are then cooled and centrifuged before analysis by LC-MS/MS.

Log peak area ratios (test compound peak area/internal standard peak area) or concentrations are plotted against incubation time and a linear fit made to the data with emphasis upon the initial rate of compound disappearance. The slope of the fit is then used to calculate the intrinsic clearance: Clint (µl min$^{-1}$ 1$^{-1}$ × 10$^6$ cells) $= -$ slope (min $- 1) \times 1,000/$[1 × 10$^6$ cells]. Data are obtained from single experiments measured with multiple time-points.

**Plasma protein binding.** Pooled and frozen plasma from selected species were obtained from commercial suppliers (The pooled and frozen plasma from human (HMPLEDTA, Lot BRH1060627) and mouse (MSEPLEDTA3-C57, Lot MSE196204) were obtained from BioreclamationIVT (NY, USA).)[64,65]. The Teflon equilibrium dialysis plate (96-well, 150 µl, half-cell capacity) and cellulose membranes (12–14 kDa molecular weight cutoff) were purchased from HT-Dialysis (Gales Ferry, CT, USA). Both biological matrix and phosphate buffer pH are adjusted to 7.4 on the day of the experiment. The reference substance is diazepam.

The determination of unbound compound is performed using a 96-well format equilibrium dialysis device with a molecular weight cut-off membrane of 12-14 kDa. The equilibrium dialysis device itself is made of Teflon to minimize non-specific binding of the test substance. Compounds are tested in cassettes of 2–5 with an initial total concentration of 1000 nM, one of the cassette compound being the positive control diazepam.

Equal volumes of matrix samples containing substances and blank dialysis buffer (Soerensen buffer at pH 7.4) are loaded into the opposite compartments of each well. The dialysis block is sealed and kept for 5 h at a temperature of 37 °C and 5% $CO_2$ environment in an incubator. After this time, equilibrium will have been reached for the majority of small molecule compounds with a molecular weight of $<600$. The seal is then removed and matrix and buffer from each dialysis is prepared for analysis by LC-MS/MS. All protein binding determinations are performed in triplicates. The integrity of membranes is tested in the HTDialysis device by determining the unbound fraction values for the positive control diazepam in each well.

At equilibrium, the unbound drug concentration in the biological matrix compartment of the equilibrium dialysis apparatus is the same as the concentration of the compound in the buffer compartment. Thus, the percent unbound fraction (fu) can be calculated by determining the compound concentrations in the buffer and matrix compartments after dialysis as follows: %fu $= 100 \times$ buffer conc after dialysis/matrix conc after dialysis. The device recovery is checked by measuring the compound concentrations in the matrix before dialysis and calculating the percent recovery (mass balance). The recovery must be within 80-120% for data acceptance.

**P-glycoprotein binding.** P-glycoprotein (permeability-glycoprotein, abbreviated as 'P-gp' also known as multidrug resistance protein 1 (MDR1)) is the most studied and best characterized drug transporter. The P-gp assay evaluates the ability of test compounds to be transported transcellularly as a P-gp substrate[66]. The assay uses transfected LLC-PK1 cells (porcine kidney epithelial cells, obtained from the Netherlands Cancer Institute) expressing human or mouse P-gp, cultured on 96-well semi-permeable filter membrane plates, where they form a polarized monolayer with tight junctions, and act as a barrier between apical and basolateral compartments. P-gp is expressed in the apical-facing membrane of the monolayer (tightness confirmed using Lucifer yellow). For substrate testing the assay determines the unidirectional permeability (Papp) of a test compound by separately dosing to the apical (for A>B Papp) and basolateral (for B>A Papp) sides of the cell monolayer (that is, donor compartments) and measuring the movement of the compound into the respective receiver compartments over a 3 h incubation at 37 °C. The effect of P-gp is measured by expressing the efflux ratio (ER) of the unidirectional A>B and B>A Papp values. The mean permeability (A>B and B>A Papp) is determined in the absence of P-gp via addition of the selective inhibitor zosuquidar at a concentration of 1 µM. The ER and mean Papp are then used to categorize compound properties regarding their degree of efflux and permeability.

**PAMPA.** PAMPA (Parallel Artificial Membrane Permeability Assay) is a method which determines the permeability of substances from a donor compartment, through a lipid-infused artificial membrane into an acceptor compartment. Read-out is a permeation coefficient Peff drug as well as test compound concentrations in donor, membrane and acceptor compartments[67].

A 96-well microtiter plate completely filled with aqueous buffer solutions (pH 7.4/ 6.5) is covered with a microtiter filterplate like a sandwich construction. The hydrophobic filter material (Durapore/Millipore; pore size 0.22–0.45 µm) of the first 48 wells (sample) of the filterplate is impregnated with a 1–20% solution of lecithin in an organic solvent (dodecane, hexadecane, 1,9-decadiene). The filter surface of the remaining 48 wells (reference) is wetted with a small volume (4–5 µl) of a 50% (v/v) methanol/buffer solution. Transport studies were started by the transfer of 100–200 µl of a 250 or 500 µM stock solution on top of the filterplate in the sample and in the reference section, respectively. In general 0.05 M TRIS, pH 7.4, or 0.05 M phosphate, pH 6.5, buffers were used. The maximum DMSO content of the stock solutions was 5%.

**Kinetic solubility.** The solubility of a test compound in phosphate buffer at pH 6.5 from evaporated DMSO compound stock solution is measured over time, resulting in the kinetic solubility of the compounds.

**Triad assay in mice using cumulative dosing procedure.** These animal experiments were performed in the lab of Prof Lichtman, Virginia Commonwealth University. Male C57BL6/J mice were purchased from Jackson Laboratories (Bar Harbor, ME, USA) and were between 10 and 12 weeks of age during treatment. The animal protocol for the triad assay was approved by the Institutional Animal Care and Use Committee at Virginia Commonwealth University in accordance with the National Institutes of Health Guide for the Care and Use of Laboratory Animals[68]. After testing was completed, all mice were humanely euthanized via $CO_2$ asphyxia, followed by rapid cervical dislocation. All studies involving animals are reported in accordance with the ARRIVE guidelines for reporting experiments involving animals[69].

Each subject (23–29 g) was assessed for catalepsy, antinociception and hypothermia as a measure of *in vivo* $CB_1$ receptor activity using a cumulative dosing procedure to examine the dose-effect relationship of each drug tested[70]. Before dosing, baseline measurements of catalepsy, antinociception and hypothermia were taken and mice were injected with the vehicle consisting of 1 part ethanol, 1 part Alkumus-620 Emulphor and 18 parts 0.9% saline. At 30 min after injection, mice were again assessed for catalepsy, antinociception and hypothermia, which required 10 min to test all mice. This process was repeated using cumulative doses of HU210 (0.03–3.0 mg kg$^{-1}$), HU308 (1–100 mg kg$^{-1}$), HU910 (1–100 mg kg$^{-1}$), and JWH133 (1–100 mg kg$^{-1}$) in half log increments, such that mice were injected every 40 min and tested 30 min after each injection until the end of the test session. Separate groups of mice ($n = 8$) were utilized for each of the four drugs. All injections were given via the intraperitoneal (i.p.) route of administration in an injection volume of 10 μl per 1 g body mass.

For the antagonism experiments, mice from cumulative dosing experiments for HU308, HU910 and JWH133 were utilized again (after a 1 week washout period), to assess whether HU910 (30 mg kg$^{-1}$) would antagonize the cannabimimetic effects of an $ED_{84}$ dose of HU210. For comparison, a separate experiment was conducted using the $CB_1$ receptor antagonist/inverse agonist rimonabant (3 mg kg$^{-1}$). Mice were evenly distributed with respect to drug history across treatment groups in each of these two experiments assessing antagonism. Baseline responses were assessed prior to injections. Subjects in each experiment received an i.p. injection of either vehicle or the appropriate receptor antagonist at 0 min, a second i.p. injection of vehicle or HU210 (1.7 mg kg$^{-1}$, which reflected the approximate $ED_{84}$ value in producing antinociception and hypothermia) at 40 min. Mice were tested for catalepsy, antinociception and hypothermia 30 min after each injection.

Data from the dose-response studies were analysed via one-way repeated measures ANOVA followed by Dunnett's *post hoc* test (versus vehicle). $ED_{50}$ and $ED_{84}$ values were determined via linear regression for drugs eliciting maximal effects for a given endpoint. Data from the antagonism experiments were analysed via two-way ANOVA in which the factors were antagonist versus vehicle and HU210 versus vehicle. *Post hoc* analysis was conducted using the Holm-Sidak test. P values of < 0.05 were defined as significant.

**In vivo pharmacokinetic studies.** Male C57BL/6J mice were obtained from CRL-F (Charles River, France), male Wistar rats were supplied by RCC Füllinsdorf and male C57BL/6JRj mice were from Janvier (France). Animals were up to 12 weeks of age. All animal studies in this section were performed at Hoffmann-La Roche and approved by the Bundesamt für Lebensmittelsicherheit und Veterinärwesen (BLV) der Schweizerischen Eidgenossenschaft. Test compounds were formulated according to respective protocols either by dissolution (i.v.) or in a glass potter until homogeneity was achieved (p.o) (formulations for p.o. administration: JWH133 and HU308 were formulated as a microsuspension in Gelatine/NaCl (7.5/0.62%) in water and HU910 was formulated as a solution in ethanol/cremophor EL/NaCl 0.9% (5/5/90%)). Formulations were injected i.v. using a 30G needle in the lateral tail vein of mice using a volume of 50 μl in the dose indicated. For p.o. applications animals were gavaged using a volume of 100 μl in the dose indicated. At the following time points blood was drawn into EDTA: 0.08, 0.25, 0.5, 1, 2, 4, 7 h (for p.o. the first time point was omitted). Six animals were used for each compound *in vivo* experiment. Animals were distributed randomly over the time course and at each time point, a volume of 100 μl of blood was taken. Quantitative plasma measurement of the compound was performed by LC-MS/MS analysis. Pharmacokinetic analysis was performed using Phoenix WinNonlin 6.4 software using a non-compartmental approach consistent with the route of administration. For assessment of the exposure Cmax, Tmax and AUC were determined from the serum concentration profiles. Parameters (CL, versus, T1/2) were estimated using nominal sampling times relative to the start of each administration. CO was extrapolated from the first concentration measured following intravenous administration.

**Data availability.** The data that support the findings of this study are included in the published article and its Supplementary Information, or are available from the corresponding author on reasonable request.

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

## Acknowledgements

This paper is dedicated to Prof Dr Raphael Mechoulam, the founder of cannabinoid research. M.S., L.H.H. and M.v.d.S. were supported by an ECHO-STIP grant from the Dutch Research Council-Chemical Sciences. A.M. was supported by the RPF program. Partial support from Ministero dell'Istruzione, dell'Università e della Ricerca (PRIN 2010–2011 grant to M.M.) is gratefully acknowledged. P.P. was supported by the Intramural Research Program of NIH/NIAAA. Mark Rau (laboratory J.G.) is acknowledged for measuring AEA uptake inhibition. Pharmacokinetics *in vivo* laboratories, Drug Disposition and Safety laboratories. Pharmaceutical Sciences pRED and F. Hoffmann-La Roche Ltd, Basel, Switzerland are acknowledged for performing the *in vitro* DMPK and *in vivo* studies. T.W.G and A.H.L were supported by NIH grants P30 DA033934 and T32DA007027.

## Author contributions

M.v.d.S., M.G., L.H.H., M.M. and P.P. conceived the research, M.S., M.v.d.S., M.G., L.H.H., M.M., U.G., J.F., C.U., M.C., V.D.M., J.G., A.H.L and P.P. designed research approach. M.S., T.W.G., F.F., L.d.P., C.U., B.R., C.P., N.v.G., D.F., C.M., A.C., M.D.G., J.S., H.d.V., N.M., L.X., G.A., M.P.B., E.D.M. and H.D. performed experiments and analysed raw data. M.S., M.G., P.P. and M.v.d.S analysed all data together and wrote the manuscript. U.G., J.F., A.M., L.H.H., M.C., V.D.M., J.G., A.H.L. and M.M. provided useful comments and feedback for the manuscript.

## Additional information

**Competing financial interests:** Industry authors U.G., J.F., C.U., B.R. and C.P. are full-time employees of Hoffmann-La Roche and P.P. is full time employee of NIH. All academic authors state that they have no conflict of interest.

