## [Peer Review File · Nature Communications]

Reviewer #2 (Remarks to the Author)

Soethoudt et al. manuscript titled "Cannabinoid CB2 receptor ligand profiling reveals biased signaling and off-target activity: Implications for drug discovery" have tried to address the contradictory results in pharmacology and in vivo data obtained on CBR ligands from different labs. Although this is a tremendous effort on part of authors to coordinate different labs to obtain reproducible and discernible data, the impact of such a study is not fully supported by the findings in the paper.

Abstract: Insert space between signal transduction (line 7).

Introduction: More clarity on how the biochemical and off-target assays utilized helps in discovery of novel compounds selective for CB2R is not well discussed.

Results: Table S1 data is reported in page 5 of results section. However the inability of SR144428 to cross the blood-brain barrier (BBB) is not discussed here, nor in the discussion section. Further ADME studies needs to be investigated to understand the results for not crossing the BBB.

Table S1: Please add in more detailed description on the physicochemical properties definitions so that the table speaks for itself.

Page 6: The authors have reported using membrane fractions of CHO cells expressing hCBRs. In addition the mouse brain and spleen as a source of mouse CBRs. However the selectivity of compounds JWH133, HU308 HU910 for CB2 is more with human receptor and selectivity for mCB2R is SR1144528 and AM630. Please explain what could be possible cross-species differences? Is it isoform expressions of CB2R in mouse spleen and brain or homology between CB1 and CB2 that results in selectivity discrepancies? Expression data of isoforms on mouse and spleen and receptor expression between the recombinant cell lines compared to tissues needs to be provided. The by itself does not explain fully the affinity results.

Figure S2. Please label each dot with a number and assign the full name of each number to the ligand described in the legend. The Figure S2 shows coorelation, but does not provide the details of the ligand corresponded in the graph.

Page 7: JWH133 was highly selective in affinity studies for human and mouse forms CB2R and is not selective in functional assay. Can the authors explain the reason for the difference? The affinity not the receptor typically governs the specificity and also the downstream signaling. Please provide functional data for G protein coupling with mouse receptors? Would JWH133 also behave like a prtail agonist and less selective for mouse CB2R?

Page 8 (first paragraph): Again the compounds JWH133, JWH015 HU910 which were partial agonist in GTP γ S assays behaved as full agonists in cAMP assay. This can be explained by amplified cAMP assay, however the expression of receptor seem to dictate the efficacy. The studies need to be done with transient transfections at different expression levels to fully characterize the agonism of each ligand.

Page 8-9: In table5S, which the authors have represented as Table IV (not clear if the numbering is wrong), have shown the JWH133, JWH015 HU910 as partial agonist for beta-arrestin. This is not accurate since the compounds were partial in their ability to couple to G protein as shown in the GTP γ S assays. The functional bias for arrestin is not correct. Please provide the effect of bias signaling also with mouse receptors to truly characterize selectivity for CB2R and bias effect. In ERK assay, the compounds JH133, HU308 and HU910 did not show any functional bias. Interestingly, for pERK the selectivity for CB2R is lost for HU910. Please explain how different the signaling cascade works for these three different ligands is given that they seem to have similar signaling properties upstream.

Does CB2R signal GIRK efficiently as CB1R? How different is the signaling? Please explain the selectivity of JWH015 to signal through G1RK/CB2R?

Page 9: Not clear how the authors came to the conclusion that HU910 and HU308 are well-balanced ligands, when the compounds showed partial agonism in b-arrestin and GTP γ S assays.

Page 11: The authors have tried to explain some of differences could also be due to off-targets effects. However HU910 and HU308 hit nine and four off-targets. This is huge and lot of data explained above including the in vivo data cannot be fully ascertained to effects on CB2R.

Reviewer #3 (Remarks to the Author)

This manuscript recapitulates a large collaborative study among different public and private institutions to evaluate the profile of the most relevant CB2 cannabinoid receptor ligands in order to reach a consensus to recommend specific CB2 agonists for preclinical studies. The manuscript is very well written and provides a large amount of information to compare the most relevant CB2 ligands. This information can be very helpful to clarify the usefulness of these ligands in preclinical studies. However, I have several important doubts about specific issues described in the manuscript and the methodology employed.

- As the authors explain, the main interest of these results is related to the possible use of these ligands in preclinical studies. However, the main interest of the CEREP panel is to identify off-target activity that can limit the clinical use of the tested compounds considering the possible link to adverse effects in humans. This panel does not evaluate multiple important targets that can represent serious bias for the preclinical studies since they are not of primary interest to detect these possible human side effects. Why a customized panel for detecting the main targets for preclinical studies was not used in these studies?
- The compounds were tested on six different TRP channels? What was the rationale to select these specific TRP channels considering that other TRP channels could also interact with these cannabinoid ligands and were not included?
- The selected CB2 ligands were tested on three of the in vivo responses included in the cannabinoid tetrad. The inclusion of the fourth cannabinoid response of this tetrad, i.e., hypolocomotion is also required to allow an appropriate comparison of the compounds.
- The use of CP55940 as a reference ligand is certainly an important bias considering its extreme potency in the cAMP assay.
- Why only three compounds were evaluated on NAPE-PLD, DAGL and MAGL overexpressing cells? The other ligands included in this work should also be tested in these assays.
- Preclinical studies have mainly provided proof-of-concept data about the efficacy of CB2 cannabinoid agonists to PREVENT the development of chronic pain. However, the clinical pain trials were designed to TREAT the manifestations of chronic pain that has already developed in patients. Therefore, it is not surprising the negative results found in these clinical trials considering their experimental design.

Minor comments

- It is really surprising the lack of selectivity of JWH133 on the GTPgammaS assay. An appropriate explanation to this unexpected result is required.
- The details about the preparation of the oral aqueous microsuspensions must be provided.

Reviewer #4 (Remarks to the Author)

This is a rigorous characterization of the physiochemical properties of 18 CBR ligands- as well as molecular pharmacological characterization. The authors conclude that the compounds are not select. The findings are not surprising, yet they may be useful to other researchers. The authors may consider publication in a journal that allows for the supplemental data they have included here to be published as primary data as the heart of the study is in the details.

Comments from Reviewers:

We would like to thank all reviewers for their excellent suggestions to improve our manuscript. Please find our responses to each comment below

Reviewer 1:

Comments to the Author

1) Soethoudt et al. manuscript titled "Cannabinoid CB₂ receptor ligand profiling reveals biased signaling and off-target activity: Implications for drug discovery" have tried to address the contradictory results in pharmacology and in vivo data obtained on CBR ligands from different labs. Although this is a tremendous effort on part of authors to coordinate different labs to obtain reproducible and discernible data, the impact of such a study is not fully supported by the findings in the paper.

The discovery of the cannabinoid CB₂ receptor, which is mainly expressed on immune cells, has raised huge hopes for the development of new anti-inflammatory, neuroprotective and analgesic drugs that could be devoid of the psychotropic actions of THC – the main component in marijuana – and other cannabinoid CB₁ receptor ligands. The controversial finding of the cannabinoid CB₂ receptor expression in the brain under normal physiological conditions has contributed to considerable delays towards this achievement (Rogers, *Nature Medicine*, 2015, 21, 966). The answer as to whether or not CB₂ receptor has clinical importance is undoubtedly linked to its alleged participation in the control of key functions in central neurons, and the suggestion of such participation is undoubtedly biased by the use of non-selective pharmacological, immunological and genetic tools. Therefore, a whole avenue of clinical development is currently being hampered by, among others, the lack of clarity of whether or not the current pharmacological tools are truly selective (See Manley *et al*, *Bioorganic & Medicinal Letters*, 2011, 21, 2359). In this frame, we believe that the work performed by our consortium on the CB₂ receptor ligands is important and will impact the field and drug discovery.

The novelty and uniqueness of our data set lie in the fact that we tested all 18 reference compounds at the same time under the same experimental conditions and repeated some of the experiments in multiple labs. This tremendously increases the data reproducibility and confidence in ligand selection. In addition the data gathered from the number of functional assays on multiple receptor orthologues is also unprecedented. This allowed our consortium to apply a functional pathway bias analyses (using the new method of Westhuizen *et al*, reported only 2015) for the first time on the CB₂ receptor ligands. This gave us insight in the functional differences not only between widely used synthetic ligands, but also for the endogenous ligands of the CB₂ receptor. This has potentially important consequences for the interpretation of the cellular and in vivo results. Furthermore, the comprehensive and extended off-target profiling has revealed previously unknown interactions of the ligands with physiologically relevant proteins. On top of that we extended our off-target profiling to an in vivo model to detect cannabimimetic activity not only of the parent molecules, but also of their potential metabolites. Finally, we performed in vivo pharmacokinetics to determine whether sufficient plasma concentrations will be reached to interact with the target. All these data together in one paper is unique and has never before been presented in such concise and comprehensive manner.

We believe that our findings will aid the cannabinoid research field to develop novel drugs, but that especially our unique approach will be applicable to any other protein target in drug discovery research.

- 2) Abstract: Insert space between signal transduction (line 7). **Corrected.**
- 3) Introduction: More clarity on how the biochemical and off-target assays utilized helps in discovery of novel compounds selective for CB₂R is not well discussed. **We have adapted the introduction.**
- 4) Results: Table S1 data is reported in page 5 of results section. However the inability of SR144428 to cross the blood-brain barrier (BBB) is not discussed here, nor in the discussion section. Further ADME studies needs to be investigated to understand the results for not crossing the BBB.

The comment about the inability of SR144528 to cross the BBB was a reference to a study that was performed previously by another group, but we think there might be a chance that a substantial amount of SR144528 does

reach the brain, based on the membrane permeation and P-Gp data. However, this is not important for the conclusion of our study. Thank you for pointing us to this unclarity: we have removed this statement.

5) Table S1: Please add in more detailed description on the physicochemical properties definitions so that the table speaks for itself. **Added.**

6) Page 6: The authors have reported using membrane fractions of CHO cells expressing hCBRs. In addition the mouse brain and spleen as a source of mouse CBRs. However the selectivity of compounds JWH133, HU308 HU910 for CB2 is more with human receptor and selectivity for mCB2R is SR1144528 and AM630. Please explain what could be possible cross-species differences? Is it isoform expressions of CB2R in mouse spleen and brain or homology between CB1 and CB2 that results in selectivity discrepancies? Expression data of isoforms on mouse and spleen and receptor expression between the recombinant cell lines compared to tissues needs to be provided. The by itself does not explain fully the affinity results.

Although there are differences in CBR expression between the recombinant cell lines and the mouse tissue, the binding affinity measured by radioligand displacement studies is independent of receptor expression and represents solely the interaction energy of the compound with the protein. The cross-species differences are, therefore, explained by differences in the primary sequences of the receptors. The consistent increase of selectivity of the antagonists on the mouse CBRs and reduced selectivity of the agonists can be explained by the fact that these ligands stabilize different receptor conformations and interact with different amino acid residues in the binding pocket. We have added this explanation to our discussion section, thanks for bringing this unclarity to our attention.

7) Figure S2. Please label each dot with a number and assign the full name of each number to the ligand described in the legend. The Figure S2 shows correlation, but does not provide the details of the ligand corresponded in the graph.

We added the numbers.

8) Page 7: JWH133 was highly selective in affinity studies for human and mouse forms CB2R and is not selective in functional assay. Can the authors explain the reason for the difference? The affinity not the receptor typically governs the specificity and also the downstream signaling. Please provide functional data for G protein coupling with mouse receptors? Would JWH133 also behave like a partial agonist and less selective for mouse CB2R?

Thank you for bringing this discrepancy to our attention. The selectivity of the ligands is reported as the ratio of EC50 (potency) values for CB2R over CB1R. The maximum effect (intrinsic efficacy) of JWH-133 was, however, not take into account. If the intrinsic efficacy of a ligand is very low, the biological activity of the ligand may not be physiologically relevant. Traditionally, a compound with an intrinsic efficacy lower than 50% @ 10 uM, is considered as "not active". We should, therefore, have designated JWH133 as "not active" in the functional CB1R assay. We have adapted our manuscript accordingly. In addition, we performed extra experiments in which we tested JWH133 on G-protein signaling on mCB₂R. JWH133 displayed the same potency and acted as a full agonist.

9) Page 8 (first paragraph): Again the compounds JWH133, JWH015 HU910 which were partial agonist in GTP_γS assays behaved as full agonists in cAMP assay. This can be explained by amplified cAMP assay, however the expression of receptor seems to dictate the efficacy. The studies need to be done with transient transfections at different expression levels to fully characterize the agonism of each ligand.

As indicated by the referee, it is actually quite common for ligands to behave as partial agonists on G-protein activation, while full agonism, due to signal amplification, is observed on downstream signaling pathways. Our results are in line with these observations. In fact, CP55940 has been extensively tested on cAMP signaling using different expression levels and is always a full agonist. Since we have found many agonists to match the activity of CP55940 in our cAMP assay, we do not think additional cAMP assays with different expression levels will provide novel information.

10) Page 8-9: In table 5S, which the authors have represented as Table IV (not clear if the numbering is wrong) have shown the JWH133, JWH015 HU910 as partial agonist for beta-arrestin. This is not accurate since the compounds were partial in their ability to couple to G protein as shown in the GTPγS assays.

The numbering of the Table has been corrected. The classical model of GPCR signaling states that beta-arrestin recruitment is dependent on G protein signaling, but more recent models show that beta-arrestin recruitment can also occur independently from G-protein signaling (in particular for biased GPCRs; see Rajagopal, S. *et al.*, "Teaching old receptors new tricks: biasing seven-transmembrane receptors", *Nat Rev Drug Discov*, 2010). Our data, therefore, support the new model that beta-arrestin signaling can be independent of G-protein signaling.

The functional bias for arrestin is not correct.

We apologize that our original statement about this analysis artefact was not clear. We argue that the observed bias signaling for all compounds towards beta-arrestin over cAMP is actually caused by a bias of the reference ligand CP55940 itself. We have clarified this point in the manuscript.

11) Please provide the effect of bias signaling also with mouse receptors to truly characterize selectivity for CB2R and bias effect.

Thank you for this valuable suggestion. We have performed additional experiments in which we characterized the G-protein activation and beta-arrestin recruitment also for mouse CB₂ and CB₁ receptors for HU308, HU910 and JWH133 – the ligands that we recommend to be used as gold standards. We performed additional biased signaling pathway analysis on the mouse receptors. Interestingly, we found that these ligands demonstrated a bias towards G-protein signaling.

12) In ERK assay, the compounds JWH133, HU308 and HU910 did not show any functional bias. Interestingly, for pERK the selectivity for CB2R is lost for HU910. Please explain how different the signaling cascade works for these three different ligands is given that they seem to have similar signaling properties upstream.

Although HU910 had a reasonably low potency in pERK signaling (pEC₅₀ = 5.5), it was a full agonist on CB2R, while on CB1R it had only 12% effect at 10 μM and a pEC₅₀ of less than 4, meaning a selectivity of HU910 of >30x for CB2. Therefore, the selectivity of HU910 is not lost in this signaling pathway.

13) Does CB2R signal GIRK efficiently as CB1R? How different is the signaling? Please explain the selectivity of JWH015 to signal through G1RK/CB2R? The results of JWH015 in the GIRK assay are actually quite consistent with the results from the other assays, so in our view GIRK signaling is not different on CB2 vs CB1.

14) Page 9: Not clear how the authors came to the conclusion that HU910 and HU308 are well-balanced ligands, when the compounds showed partial agonism in b-arrestin and GTPγS assays.

Our conclusion is based on the results of the operational model of Black and Leff (1983) and the analysis of Westhuizen *et al.*, 2015, which calculate signal transduction strength on a given pathway, taking into account a) the maximal effect of the system used, b) the agonist concentration, c) the agonist's maximum efficacy, d) the ligand affinity for the receptor and e) the transducer slope. From this analysis it becomes clear that HU910 and HU308 do not display statistical significant bias, therefore it is likely that the partial agonism observed with these compounds is a result of the experimental system.

15) Page 11: The authors have tried to explain some of differences could also be due to off-targets effects. However HU910 and HU308 hit nine and four off-targets. This is huge and lot of data explained above including the in vivo data cannot be fully ascertained to effects on CB2R.

The CEREP data is generated from single concentration (10 μM) binding assays with human orthologues with the aim to predict adverse effects in in vivo studies. The observed off-targets are unlikely to have influenced the in vitro molecular pharmacology studies, because they have mostly been performed with transfected cell lines overexpressing CB2R or CB1R. We did not see any effect of the ligands in control cell lines without overexpressing the CB2 or CB1R (data not shown). In addition, we determined full dose-response curves for HU910 on all 9 off-targets: IC₅₀ > 10 μM were found and only the dopamine uptake reporter displayed an IC₅₀ of 1.40E-06 M. In conclusion, we do not think there is that polypharmacology has influenced our results at physiologically relevant concentrations.

Reviewer 2:

Comments to the Author

1) This manuscript recapitulates a large collaborative study among different public and private institutions to evaluate the profile of the most relevant CB2 cannabinoid receptor ligands in order to reach a consensus to recommend specific CB2 agonists for preclinical studies. The manuscript is very well written and provides a large amount of information to compare the most relevant CB2 ligands. This information can be very helpful to clarify the usefulness of these ligands in preclinical studies. However, I have several important doubts about specific issues described in the manuscript and the methodology employed. As the authors explain, the main interest of these results is related to the possible use of these ligands in preclinical studies. However, the main interest of the CEREP panel is to identify off-target activity that can limit the clinical use of the tested compounds considering the possible link to adverse effects in humans. This panel does not evaluate multiple important targets that can represent serious bias for the preclinical studies since they are not of primary interest to detect these possible human side effects. Why a customized panel for detecting the main targets for preclinical studies was not used in these studies?

We have selected the CEREP panel of protein targets that are most known and most likely to be targeted in humans and also most likely to cause major problems. With this we aimed to identify rapid feedback on the promiscuity and potential drug safety aspects associated with structural features of the molecules (Adverse drug reactions (ADRs) are generally dose-dependent and can be predicted from the pharmacological profile of the candidate compound). To design our panel an extensive analysis of historical screening data was done and we observed: (1) some ligand binding interactions are predictive of others; for instance, if a compound binds to the 5-HT_{2B} receptor, it will likely also interact at histamine H1 and α 1-adrenergic receptors; and (2) some targets are almost never hit. Furthermore additional targets, based on low redundancy were added to the already created set. This gave a panel of 51 targets, which covered > 95% of all hits in the BioPrint™ dataset, which is a commercial dataset of drug-like compounds screened against 159 safety-relevant targets. Additional information on reducing safety-related drug attrition: the use of *in vitro* pharmacological profiling can be found in this Nature review (Bowes et al., 2012): <http://www.nature.com/nrd/journal/v11/n12/pdf/nrd3845.pdf>. To prioritize compounds, it should suffice to screen them at a reduced set of predictive targets, and to omit rarely hit sites. Therefore, our panel is a very good selection of ADR relevant targets (the panel was adjusted over time and therefore we don't have the exact data for each and every compound).

In addition, we have made a customized panel of protein targets to test our reference ligands on protein targets that were most likely to bias any effects mediated by the CB₂R (i.e. proteins of the endocannabinoid system, TRP channels). The proteins we tested the ligands on were carefully selected and many thoughts were put into it. Therefore, we think that testing the ligands on even more proteins would not be necessary.

2) The compounds were tested on six different TRP channels? What was the rationale to select these specific TRP channels considering that other TRP channels could also interact with these cannabinoid ligands and were not included?

To our knowledge, only a subset of TRP channels has been shown (to date) to be activated by cannabinoid ligands, and these are the TRPV/TRPA1/TRPM8 ones, which have been of great interest because of their expression in sensory neurons (and brain), and thus obvious (and validated) targets for analgesics. These specific TRP channels are also considered to be thermosensitive and therefore possibly related with endocannabinoid actions, and for this reason we tested cannabinoids thereupon.

3) The selected CB2 ligands were tested on three of the *in vivo* responses included in the cannabinoid tetrad. The inclusion of the fourth cannabinoid response of this tetrad, i.e., hypolocomotion is also required to allow an appropriate comparison of the compounds.

Thank you for your feedback. The reason why the hypolocomotion is not included in this particular study, is that we used a cumulative dose-response treatment of the mice, which does not allow to measure the effects on hypolocomotion of these compounds, because the repeated testing of the mice leads to acclimation to the locomotor activity chambers, which leads in turn to a biased outcome. The cannabinoid tetrad assay has been set up as a vast approach to prove specifically the effects of the CB₁ receptor *in vivo*. We have used three out of four assays on the cannabinoid tetrad to exclude activity of CB₂R agonists on the CB₁R. Therefore we think that three out of four is enough to exclude activity on CB₁R, whereas the fourth might be essential to prove CB₁ receptor action.

4) The use of CP55940 as a reference ligand is certainly an important bias considering its extreme potency in the cAMP assay.

We agree, we have adjusted the explanation in the main text, thanks for pointing this out.

5) Why only three compounds were evaluated on NAPE-PLD, DAGL and MAGL overexpressing cells? The other ligands included in this work should also be tested in these assays.

Thank you for this suggestion. We have added the data as requested.

6) Preclinical studies have mainly provided proof-of-concept data about the efficacy of CB₂ cannabinoid agonists to PREVENT the development of chronic pain. However, the clinical pain trials were designed to TREAT the manifestations of chronic pain that has already developed in patients. Therefore, it is not surprising the negative results found in these clinical trials considering their experimental design.

It is true that there were many studies that were flawed in their experimental design, causing CB₂R agonists to be found inactive or not able to validate CB₂R as a target. This is actually also one of the reasons we have conducted this study, so we have added this to our introduction.

Minor comments

7) It is really surprising the lack of selectivity of JWH133 on the GTPgammaS assay. An appropriate explanation to this unexpected result is required.

Thank you for bringing this discrepancy to our attention. The selectivity of the ligands is reported as the ratio of EC₅₀ (potency) values for CB₂R over CB₁R. The maximum effect (intrinsic efficacy) of JWH-133 was, however, not taken into account. If the intrinsic efficacy of a ligand is very low, the biological activity of the ligand may not be physiologically relevant. Traditionally, a compound with an intrinsic efficacy lower than 50% @ 10 uM, is considered as "not active". We should, therefore, have designated JWH133 as "not active" in the functional CB₁R assay. We have adapted our manuscript accordingly. In addition, we performed extra experiments in which we tested JWH133 on G-protein signaling on mCB₂R. JWH133 displayed the same potency and acted as a full agonist.

8) The details about the preparation of the oral aqueous microsuspensions must be provided.

We have added them as requested, thanks for bringing this to our attention.

Reviewer 3:

Remarks to the Author

This is a rigorous characterization of the physicochemical properties of 18 CBR ligands- as well as molecular pharmacological characterization. The authors conclude that the compounds are not select. The findings are not surprising, yet they may be useful to other researchers. The authors may consider publication in a journal that allows for the supplemental data they have included here to be published as primary data as the heart of the study is in the details.

We thank the reviewer for his/her compliments. Perhaps, we understated the uniqueness of our approach and the resulting findings. Since our goal was to identify the gold standard ligands for CB₂ receptor pharmacology, it is not surprising that binding and some functional data of the reference compounds are already present in the literature. These data are, however, scattered across many different publications in which varying and sometimes contradictory results have been reported - very likely due to varying experimental conditions. This difficulty to compare and interpret the results of these ligands has led to confusion in the literature about which ligands to use for in vivo experiments to validate the CB₂ receptor as a therapeutic target. This has wasted a lot of resources (time and money-wise) and led to the unnecessary use of animals. Unfortunately, this situation is not unique for the CB₂ receptor field, but happens regularly also in other research fields. The US National Institutes of Health (NIH) also shares the concerns from many scientists about the reproducibility issues in biomedical research and is

demanding action to counter this severe problem (See Collins and Tabak, Nature, 2014,505, 612). We stepped up to this challenge and we envision that our manuscript will improve reproducibility of CB₂ receptor pharmacological studies and our approach may impact other research fields as well.

We believe that our findings will aid the cannabinoid research field, but that especially our unique approach will be applicable to any other protein target in drug discovery research. (See also our comments to point 1 of reviewer 1)

We feel that Nature Communications as an **online open access** journal is a perfectly suited journal to provide the research community and the general public access to all our data.

Reviewer #2 (Remarks to the Author)

Thanks for responding to the questions.

Reviewer #3 (Remarks to the Author)

The authors have changed the manuscript taking into consideration several concerns raised by the reviewers and have included additional data in the tables. However, several concerns still remain open:

As the author state in the responses to the reviewers comments, the main current limitation for advancing in the understanding of the physiopathological role of the CB2 cannabinoid receptor is the lack of clear knowledge about the expression of this receptor in the brain under normal physiological conditions. The absence of selective immunological tools is the main reason for this lack of knowledge. However, the results of the present manuscript with an extensive characterization of the most widely used CB2 ligands would not provide important advances for using these pharmacological tools to resolve this crucial question that still remains open.

The use of the CEREB panel instead of a customized panel appropriate for testing CB2 cannabinoid ligands is not well justified. As the authors explain in their answer, this panel has been designed to identify targets related to adverse drug reactions, but not for identifying possible relevant pharmacological targets that could interfere with CB2 cannabinoid ligands.

The fourth response of the cannabinoid tetrad, hypolocomotion, can be evaluated on this experimental design using an appropriate acclimation to the locomotor activity chambers allowing a homogeneous locomotor response of the animals. The absence of this important response closely related to the pharmacological responses of the cannabinoid ligands represents a limitation that must be clearly discussed in the manuscript.

Comments from Reviewers:

We thank the reviewers for their time and valuable comments to improve our manuscript.

Reviewer 2:

Comments to the Author

Thanks for responding to the questions.

You are welcome.

Reviewer 3:

Comments to the Author

The authors have changed the manuscript taking into consideration several concerns raised by the reviewers and have included additional data in the tables. However, several concerns still remain open: As the author state in the responses to the reviewers comments, the main current limitation for advancing in the understanding of the physiopathological role of the CB₂ cannabinoid receptor is the lack of clear knowledge about the expression of this receptor in the brain under normal physiological conditions. The absence of selective immunological tools is the main reason for this lack of knowledge. However, the results of the present manuscript with an extensive characterization of the most widely used CB₂ ligands would not provide important advances for using these pharmacological tools to resolve this crucial question that still remains open.

We agree with the reviewer that the lack of specific CB₂ receptor antibodies impairs the pinpointing of the cellular expression pattern of this GPCR, especially in the brain. Therefore, most studies use chemical tools to address CB₂ receptor function in the central nervous system. Since, the widely used reference compounds are not well characterized, especially in relation to the cannabinoid CB₁ receptor (across different species and in vivo), the conclusions of in vivo studies with these CB₂ receptor ligands reported in the literature are confounded. For example, see Manley *et al*, *Bioorganic & Medicinal Letters*, 2011, 21, 2359. This was exactly one of the reasons to initiate our large multinational and multidisciplinary collaboration between academia and industry. Our extensive studies have allowed us to propose JWH-133, HU-910 and HU-308 as the gold standards to study cannabinoid CB₂ receptor biology and for target validation purposes.

The use of the CEREP panel instead of a customized panel appropriate for testing CB₂ cannabinoid ligands is not well justified. As the authors explain in their answer, this panel has been designed to identify targets related to adverse drug reactions, but not for identifying possible relevant pharmacological targets that could interfere with CB₂ cannabinoid ligands.

The reviewers' criticism that we did not use a customized panel appropriate for testing CB₂ receptor ligands seems to reflect a misreading of our manuscript. Next to the CEREP panel, we have specifically compiled an off-target panel consisting of 18 proteins, which are known to interact with cannabinoid CB₂ receptor ligands and/or the endocannabinoids. The panel consists of the following proteins:

G-protein coupled receptors known to bind cannabinoids:

- human and mouse CB₁ receptor
- GPR55

Proteins involved in biosynthesis and metabolism of endocannabinoids:

- Diacylglycerol lipase-alpha

- *N*-acylphosphatidylethanolamine phospholipase D,
- monoacylglycerol lipase
- α,β -hydrolase domain containing protein 6
- α,β -hydrolase domain containing protein12,
- cyclooxygenase-2,
- fatty acid amide hydrolase
- endocannabinoid transporter
- general serine hydrolase activity in mouse brain proteome using activity-based protein profiling

Ion channels known to interact with cannabinoids

- Transient Receptor Potential Vanilloid type 1
- Transient Receptor Potential Vanilloid type 2
- Transient Receptor Potential Vanilloid type 3
- Transient Receptor Potential Vanilloid type 4
- Transient Receptor Potential A1
- Transient Receptor Potential M8

The fourth response of the cannabinoid tetrad, hypolocomotion, can be evaluated on this experimental design using an appropriate acclimation to the locomotor activity chambers allowing a homogeneous locomotor response of the animals. The absence of this important response closely related to the pharmacological responses of the cannabinoid ligands represents a limitation that must be clearly discussed in the manuscript.

We have included the following section:

“Lastly, HU308, HU910 and JWH133 were found to have no CB₁ activity in vivo when tested in the mouse cannabinoid triad (anti-nociception, catalepsy and hypothermia). The fourth assay of the mouse cannabinoid tetrad, hypolocomotion, was not included in this study, because the mice acclimatized to the locomotor activity chamber due to cumulative dosing procedure. The omission of the hypolocomotor assay does not confound our conclusions, because to classify a compound as an in vivo active CB₁ receptor agonist, the ligand must be active in all four assays. Since HU308, HU910 and JWH133 did not elicit activity in the first three assays, the compounds are not active on CB₁ receptor in vivo.”